# TDP-1, the *Caenorhabditis elegans* ortholog of TDP-43, limits the accumulation of double-stranded RNA

Tassa K Saldi[1,*], Peter EA Ash[2], Gavin Wilson[3,4], Patrick Gonzales[5], Alfonso Garrido-Lecca[1], Christine M Roberts[5], Vishantie Dostal[5], Tania F Gendron[2], Lincoln D Stein[3], Thomas Blumenthal[1], Leonard Petrucelli[2] & Christopher D Link[5,6]

## Abstract

*Caenorhabditis elegans* mutants deleted for TDP-1, an ortholog of the neurodegeneration-associated RNA-binding protein TDP-43, display only mild phenotypes. Nevertheless, transcriptome sequencing revealed that many RNAs were altered in accumulation and/or processing in the mutant. Analysis of these transcriptional abnormalities demonstrates that a primary function of TDP-1 is to limit formation or stability of double-stranded RNA. Specifically, we found that deletion of *tdp-1*: (1) preferentially alters the accumulation of RNAs with inherent double-stranded structure (dsRNA); (2) increases the accumulation of nuclear dsRNA foci; (3) enhances the frequency of adenosine-to-inosine RNA editing; and (4) dramatically increases the amount of transcripts immunoprecipitable with a dsRNA-specific antibody, including intronic sequences, RNAs with antisense overlap to another transcript, and transposons. We also show that TDP-43 knockdown in human cells results in accumulation of dsRNA, indicating that suppression of dsRNA is a conserved function of TDP-43 in mammals. Altered accumulation of structured RNA may account for some of the previously described molecular phenotypes (e.g., altered splicing) resulting from reduction of TDP-43 function.

**Keywords** neurodegeneration; RNA editing; RNA structure; splicing
**Subject Categories** RNA Biology
The EMBO Journal (2014) 33: 2947–2966

## Introduction

TAR-DNA binding protein 43 (TDP-43) was originally discovered as a negative regulator of HIV DNA transcription (Ou *et al*, 1995).

Subsequently, TDP-43 was identified as a major component of aggregates in a variety of degenerative neurological conditions including amyotrophic lateral sclerosis (ALS) and frontotemporal lobar dementia (FTLD) (Chen-Plotkin *et al*, 2010). In affected tissues, TDP-43 is disrupted, with the formation of large, insoluble cytoplasmic aggregates and concurrent nuclear depletion. While mutations in TDP-43 have been identified in a small number of familial cases of ALS (Pesiridis *et al*, 2009), mislocalized wild-type TDP-43 appears in the vast majority of sporadic ALS and FTLD-affected tissue and in a significant percentage of other neurological disorders. These observations suggest that the wild-type function of TDP-43 is central to the disease cascade.

TDP-43 is a ubiquitously expressed RNA/DNA-binding protein containing two RNA Recognition Motifs (RRMs) that allow binding to single-stranded RNA as well as single- and double-stranded DNA (Ayala *et al*, 2005). TDP-43 proteins preferentially associate with $(UG)_n$ repeats in RNA, but this sequence is neither necessary nor sufficient for TDP-43 association *in vivo* (Buratti & Baralle, 2001; Polymenidou et al, 2011). TDP-43 is involved in many RNA-related processes including transcription, pre-mRNA splicing, mRNA stability, and micro-RNA biogenesis (Da Cruz & Cleveland, 2011). Several recent studies have characterized neuronal transcripts affected by TDP-43 (Polymenidou *et al*, 2011; Tollervey *et al*, 2011). These studies indicate that TDP-43 preferentially binds long intronic regions in pre-mRNA and maintains these transcripts by an unknown mechanism. Alternative splicing analysis in several TDP-43 knockdown models indicates TDP-43 functions to ensure correct alternative splicing of pre-mRNAs (Polymenidou *et al*, 2011; Hazelett *et al*, 2012). However, changes in splicing are not observed in the majority of transcripts with decreased levels. Therefore, the reason TDP-43 is required to maintain the levels of certain transcripts is unknown.

Knockout of TDP-43 in most model organisms is lethal or results in severe phenotypes (Feiguin *et al*, 2009; Sephton *et al*, 2010).

1   Department of Molecular, Cellular, and Developmental Biology, University of Colorado, Boulder, CO, USA
2   Department of Neuroscience, Mayo Clinic, Jacksonville, FL, USA
3   Department of Molecular Genetics, University of Toronto, Toronto, ON, Canada
4   Informatics and Biocomputing Platform, Ontario Institute for Cancer Research, Toronto, ON, Canada
5   Institute for Behavioral Genetics, University of Colorado, Boulder, CO, USA
6   Integrative Physiology, University of Colorado, Boulder, CO, USA
    *Corresponding author. Tel: +1 303 492 0340; E-mail: tassa.saldi@colorado.edu

Surprisingly, complete deletion of the *C. elegans* ortholog, TDP-1, causes minor defects (Zhang *et al*, 2012). Despite the lack of severe phenotypes in the *tdp-1* mutant, TDP-1 has molecular properties similar to its mammalian homolog. TDP-1 binds the canonical TDP-43 binding sequence [$(UG)_n$] with high affinity (Ayala *et al*, 2005), and we have shown that TDP-1 can substitute for human TDP-43 in *in vivo* splicing assays (Ash *et al*, 2010). These results imply that while *tdp-1* loss of function may be less consequential in the worm, TDP-1's basic molecular roles are likely conserved.

In this study, we investigated the role of TDP-1 on the transcriptome. We discovered that TDP-1 functions to maintain the amount of mature RNA transcripts originating from potentially double-stranded precursor RNAs and to limit nuclear dsRNA accumulation in multiple tissues. Immunoprecipitation using a dsRNA-specific antibody revealed that *tdp-1* mutant animals accumulate a variety of double-stranded transcripts indicating a global effect of TDP-1 on RNA structure or stability. Analysis of TDP-1 binding by deep sequencing of anti-TDP-1 chromatin immunoprecipitation (ChIP) indicated that TDP-1 associates with highly structured regions co-transcriptionally. The reduction of dsRNA accumulation is likely to be a conserved function of TDP-43 proteins because knockdown of mammalian TDP-43 in HeLa cells and M17 neuronal cells also causes dsRNA accumulation.

# Results

### RNA transcripts aberrantly represented in *tdp-1(ok803)*

To determine TDP-1's effect on the transcriptome, we deep-sequenced RNA from *tdp-1* mutant animals to identify changes in RNA metabolism or abundance. While two deletion alleles exist for *tdp-1*, *tdp-1(ok803)* and *tdp-1(ok781)*, only the *tdp-1(ok803)* allele appeared to be a clear null (Supplementary Fig S1B and C); therefore, our study is focused on this allele. We created and sequenced poly(A)-selected cDNA libraries from wild-type and *tdp-1(ok803)* animals. Mapping of sequenced reads showed over 50% of all ampli-fiable annotated genes were well represented (roughly 14,000 tran-scripts). A differential gene expression comparison (RPKMs) between *tdp-1(ok803)* mutants and wild-type revealed over 1,700 transcripts aberrantly represented with close to an equal number of transcripts underrepresented as overrepresented (Fig 1A; Supple-mentary Table S1). qRT–PCR verification of a select set of these abundance changes is shown in Supplementary Fig S2. Gene ontol-ogy analysis of aberrantly represented transcripts in *tdp-1* mutants indicated very few pathways were enriched among over- and under-expressed genes (Supplementary Table S2). Most enriched pathways were associated with developmental processes and stage-specific molting, which may be an artifact due to the mild growth delay reported in *tdp-1* mutant animals (Zhang *et al*, 2012).

Because TDP-43 functions in alternative splicing, we also asked whether transcripts with altered abundance in the mutant had splic-ing abnormalities. Splicing analysis identified ~350 genes with significant ($P < 0.001$) changes in splice site representation (Supple-mentary Table S3, independent verification in Supplementary Fig S3); however, the majority of transcripts altered in splicing were not altered in abundance compared to wild-type. While splicing abnor-malities may contribute to *tdp-1* loss of function defects, splicing

differences do not readily explain changes in transcript abundance in *tdp-1(ok803)* animals.

### TDP-1 regulates genes with inherent double-stranded structure

Mammalian TDP-43 maintains the abundance of transcripts with long introns. One interpretation of this observation is that tran-scripts require TDP-43 because of an inherent characteristic of the RNA transcript itself and not due to its involvement in a common functional pathway. To look for common features in RNA tran-scripts affected by TDP-1, we examined the most dramatically altered transcripts in the *tdp-1* deletion using the IGV genome browser and online databases (wormbase, aceview) for common characteristics within the RNA molecule. Interestingly, we noticed that a large proportion of altered transcripts had potential double-stranded structure. Specifically, many over/underexpressed tran-scripts contained either antisense overlap with another gene or multiple inverted repeats within intronic regions.

To quantify these effects, we first analyzed the percentage of altered genes with antisense overlap to another coding gene. While about 8% of the worm genome is arranged antisense to another gene (Thierry-Mieg & Thierry-Mieg, 2006), approximately 35% of increased and decreased transcripts were arranged in this manner, representing a highly significant enrichment ($P = 5.3 \times 10^{-181}$, hyper-geometric distribution (hgd)) (Fig 1B). We also determined the percentage of genes containing intronic inverted repeats that were aberrantly represented in *tdp-1* deletion. We limited our analysis to inverted repeats contained within introns greater than 1 kb. While about 25% of all expressed genes contained an intron inverted repeat (2,641), 40% of underrepresented genes in *tdp-1(ok803)* RNA-seq contained inverted repeats, representing a significant enrichment ($P = 4 \times 10^{-5}$, hgd, $P = 3.2 \times 10^{-3}$, chi-square test) (Fig 1B). Among overexpressed transcripts, inverted repeat containing introns were not significantly enriched, suggesting that intronic RNA structure in *tdp-1* mutants results in reduction of mRNA abundance.

Transcripts with antisense overlap to another RNA and tran-scripts containing inverted repeats are capable of forming either inter- or intra-molecular dsRNA. Because deletion of *tdp-1* perturbed the abundance of these transcripts, we hypothesized that TDP-1 has a fundamental function in the formation, or metabolism, of dsRNA.

### TDP-1 limits the accumulation of double-stranded RNA

To address the hypothesis that TDP-1 affects dsRNA metabolism, we looked directly at the amount and localization of dsRNA in *tdp-1 (ok803)* mutant worms by immunostaining with a dsRNA-specific antibody, J2. The J2 antibody is specific to dsRNA duplexes greater than 40 bp (Schonborn *et al*, 1991) and was used previously to visualize double-stranded RNA *in vivo* (Kaneko *et al*, 2011). We stained populations of wild-type and *tdp-1(ok803)* worms with the J2 antibody. As shown in Fig 2, *tdp-1(ok803)* worms had a signifi-cant increase in nuclear inclusions reactive with the J2 antibody. This increase was observed in multiple tissues including: gut (Fig 2A), muscle, and the anterior head region (Supplementary Fig S4). Importantly, detection of these inclusions was ablated by pretreatment with the dsRNA-specific RNAse V1, but not by treat-ment with the ssRNA-specific RNAse T1 (Fig 2A, bottom two panels). To confirm the specificity of the J2 antibody in worms, we

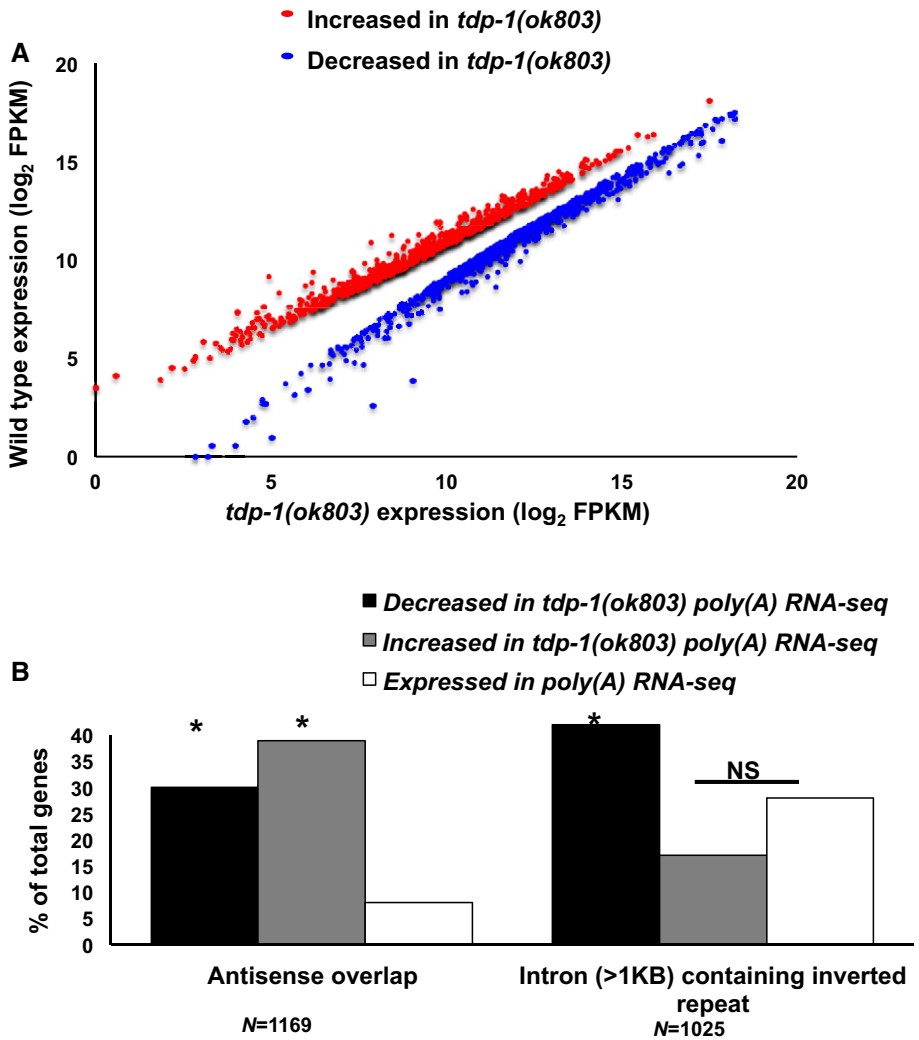

**Figure 1. TDP-1 maintains transcripts with potential dsRNA structure.**

A  Scatter plot comparing the $\log_2$ of transcript RPKMs (Reads Per Kilobase of transcript per Million mapped reads) between wild-type and *tdp-1(ok803)* poly(A)-selected RNA-seq experiments. Only transcripts significantly (corrected $P < 0.05$) increased (red) or decreased (blue) compared to wild-type are shown. Scatter plot represents significantly different genes calculated from two independent biological replicates of both wild-type and *tdp-1(ok803)* poly(A) RNA-seq prepared from L4 animals.

B  Percentage increased and decreased transcripts in *tdp-1(ok803)* poly(A) RNA-seq that have antisense overlap with another spliced gene or have intronic sequences (> 1 kb) that contain inverted repeats. $*P < 1 \times 10^{-4}$ (hypergeometric distribution); NS, not significant relative to control gene sets.

also immunostained a transgenic strain expressing a dsRNA-binding domain fused GFP (Supplementary Fig S5). J2 and GFP loci tightly overlapped, indicating they likely recognize the same substrates.

### TDP-1 is required to maintain normally low levels of A-to-I RNA editing

Since nuclear dsRNA is limited by the activity of A-to-I dsRNA-specific deaminases (ADARs) (Bass, 1998), we hypothesized that TDP-1 could function in promoting A-to-I RNA editing. To address this possibility, we investigated the amount of A-to-I editing in *tdp-1 (ok803)* mutants. In order to determine whether identified A-to-I edits represented true physiological targets, we sequenced RNA in parallel from a strain deleted for the only functional worm ADAR, *adr-2(gv42)*. Using a compiled list of all nucleotide positions that showed a discrepancy between the RNA and DNA sequence (see

Materials and Methods), we hand-annotated 210 regions containing convincing evidence of A-to-I RNA editing (Supplementary Table S4). Regions were only considered edited if A-to-I conversions were observed (by inspection of sequencing reads in IGV genome browser) in all three independent biological replicates and not observed in RNA-seq from *adr-2(gv42)* mutants (examples of representative regions scored as positive for editing are shown in Supplementary Fig S6). The majority (> 90%) of editing events occurred in introns and 3′ UTRs, and all observed editing was consistent with promiscuous editing, which occurs in long dsRNA (average of 123 positions targeted per region, see Supplementary Table S4). We found no reproducible evidence of exonic A-to-I editing. While our list is likely not an exhaustive list of all edited transcripts in *C. elegans* (only larval stage 4 was analyzed and the read depth of many intronic regions was too low to assess editing), our analysis did successfully identify eight of the ten published transcripts

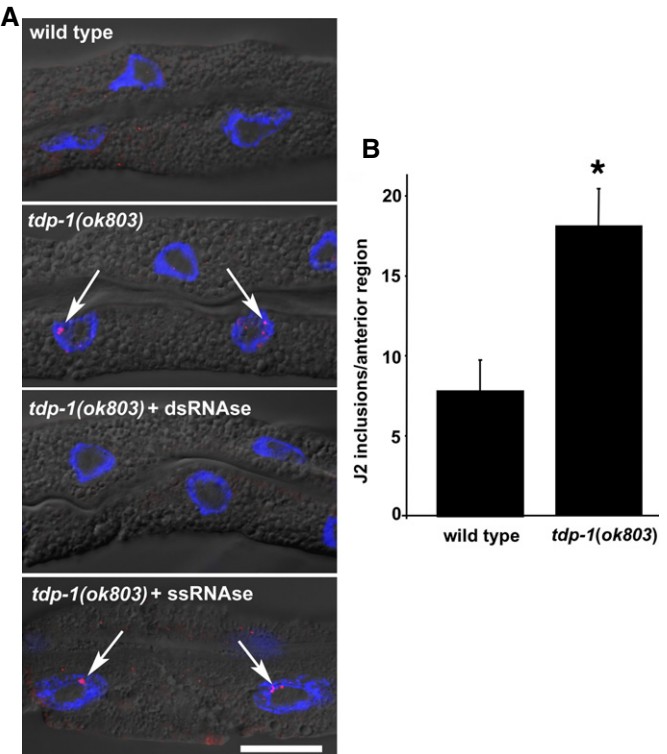

**Figure 2. TDP-1 limits the amount of nuclear dsRNA.**

A  Fixed, isolated intestinal tissue probed with anti-dsRNA antibody (J2). The J2 antibody recognizes dsRNA stretches of 40 bp or more in a sequence-independent manner. Intensely stained inclusions (red dots, indicated by arrows) were detected in intestinal nuclei (blue, DAPI counterstain) specifically in *tdp-1(ok803)* mutant worms (middle panel). J2-reactive inclusions were observed in 28% *tdp-1(ok803)* intestinal nuclei scored (30/107), but not detected in wild-type controls (0/122) or *tdp-1(ok803)* (0/105) fixed tissue pretreated with dsRNA nuclease (V1) before J2 staining. J2-reactive foci were still observed (arrows, bottom panel) in intestinal nuclei pretreated with the ssRNA-specific nuclease (T1) (29%, 12/41). Scale bar, 20 μm.

B  Quantification of anterior J2 inclusions using ImageJ software. *tdp-1(ok803)* worms had significantly more J2 inclusions (*P < 0.01, Student's *t*-test, error bars = SEM). Representative projection images used to generate this data are shown in Supplementary Fig S4.

characterized as edited due to preferential digestion with T1 RNAse (Morse *et al*, 2002) as well as 83 of the regions identified by Wu *et al* (2011). To confirm that regions positive for A-to-I editing contained sequence motifs capable of dsRNA formation, we assayed for the presence of inverted repeat/loop sequences within identified editing targets (see Materials and Methods). We found that almost all (> 98%) of the regions identified in this study contained two or more inverted repeat sequences and/or overlapped with transposons/retrotransposons (Supplementary Table S5). Transcripts containing both inverted repeats and transposons can form intra-molecular dsRNA and are known targets of A-to-I editing *in vivo* (Nishikura, 2010). These results indicate that our analysis identified editing only within potentially structured RNA sequences, which are suitable targets of worm ADAR.

To determine whether *tdp-1* maintains A-to-I RNA editing within these regions, we calculated the percent editing across all identified

edited regions that were well represented (> 20 reads, 153/210 regions) in both wild-type and *tdp-1* mutants. Roughly 50% of well-represented regions showed significantly (*P* < 0.05) altered editing (Fig 3A), and in almost all (> 85%) cases, editing was increased in the *tdp-1* deletion transcripts compared to wild-type. The most common hyper-editing occurred in intronic regions. Two examples of increased intronic editing in *tdp-1(ok803)* RNA-seq are shown (Fig 3B). Examination of transcript abundance and splicing of the worm ADAR genes revealed no substantial change between wild-type and *tdp-1* mutants (Supplementary Fig S7A). While there is no available antibody for worm ADR-2, we observed no change in ADR-1 protein levels between wild-type and mutant (Supplementary Fig S7B), suggesting that *tdp-1* does not limit A-to-I RNA editing by controlling ADAR function. Although the majority of editing increases in *tdp-1* mutants were mild (average fold increase ~1.4-fold), increased amounts of A-to-I editing indicate that the accumulation of dsRNA in *tdp-1(ok803)* mutant worms is not due to a reduction in RNA editing. Rather, increases in the structure or stability of dsRNA could result in a slightly higher frequency of editing in *tdp-1* mutants.

**Immunoprecipitation with an anti-dsRNA antibody recovers structured transcripts**

The J2 antibody can be used to immunoprecipitate dsRNA, as shown for Alu transcripts in mammalian cell lysates (Kaneko *et al*, 2011). Therefore, we analyzed J2 immunoprecipitates from both wild-type and mutant whole-worm lysates (in triplicate) to characterize the specific dsRNAs that accumulate in *tdp-1(ok803)* mutants. Because J2 immunoprecipitation was done in untreated lysate, entire transcripts containing 40 bp or more of dsRNA anywhere within the molecule will be precipitated. J2-immunoprecipitated RNA and total RNA were sequenced in a strand-specific manner. Pairwise correlation of J2-IP RNA-seq replicates indicated that the J2-IP replicates were highly similar (Supplementary Fig S8).

First, we analyzed the wild-type RNA-seq data to validate J2 immunoprecipitation (J2-IP) as an effective method to isolate dsRNA. Comparison between input and J2-immunoprecipitated RNA confirmed that J2-IP selected for a subset of transcripts (Fig 4A). As a positive control, we calculated the fold enrichment (over input) of J2-IP RNA-seq for all expressed repetitive regions (Fig 4B). Consistent with J2-IP selecting for dsRNA, we observed a fourfold to fivefold increase in expression level (RPKMs) of annotated transposon/retrotransposon sequences, as well as a twofold to fourfold increase in the expression level (RPKMs) of tandem/inverted repeats. We did not see an overall increase in the enrichment of exon sequences, confirming J2-IP is specific for transcripts originating from regions known to produce dsRNA. Examples of an enriched transposon and retrotransposon are depicted in Fig 4C. We verified the specificity of J2-IP for dsRNA transposons as this enrichment was abolished in lysates pretreated with dsRNA-specific RNase (Supplementary Fig S9).

To ask which gene transcripts were enriched by J2-IP, we compared normalized read counts between input RNA-seq and J2-IP RNA-seq for each gene. Only genes that were significantly increased (*P* < 0.05, FDR < 0.1, > 1.5-fold) over input were considered

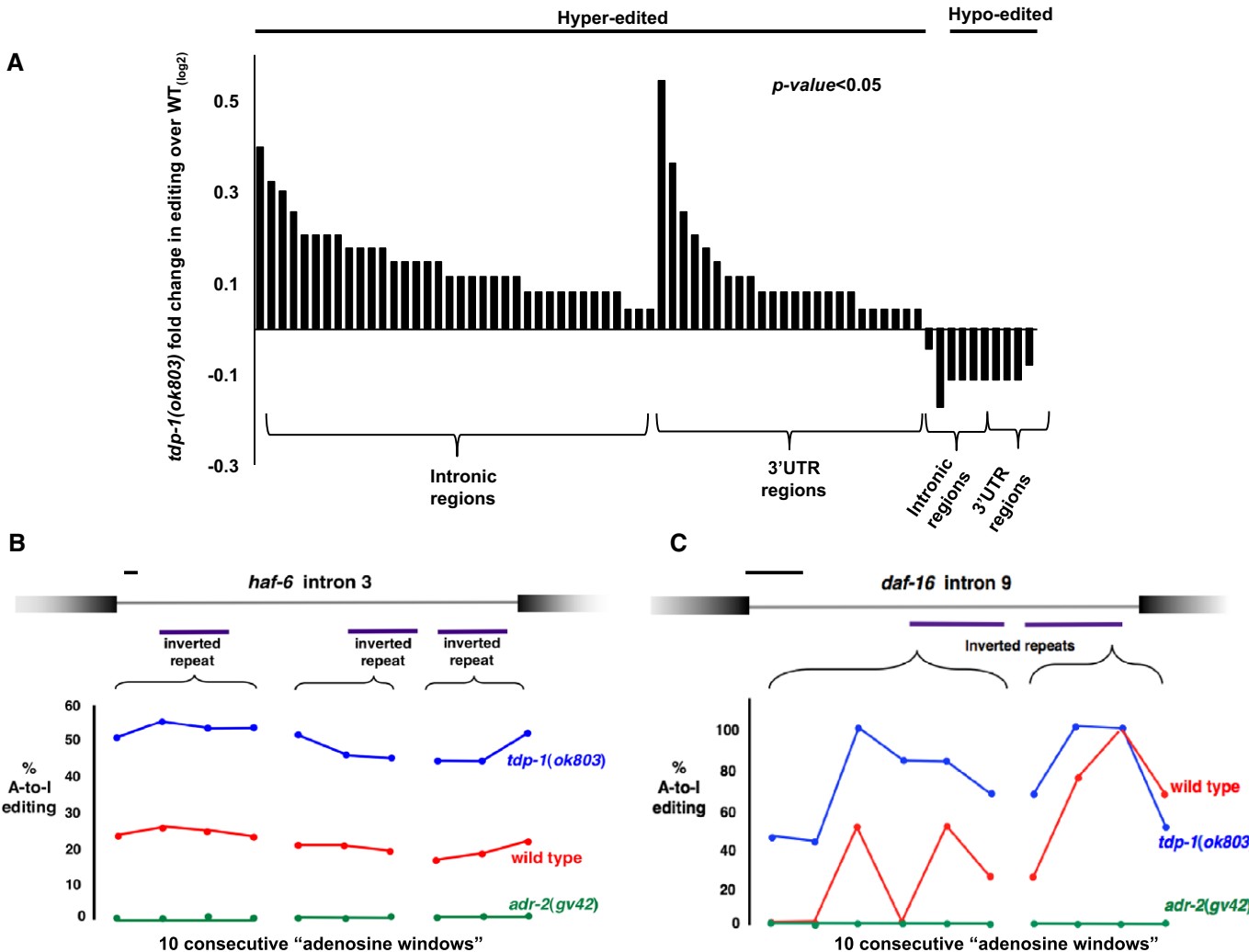

**Figure 3.  TDP-1 limits levels of A-I RNA editing.**

A    Fold change in % A-to-I editing in *tdp-1(ok803)* compared to wild-type RNA-seq is shown. For all regions, the average percent editing for each potentially edited nucleotide was calculated. Bars in graph represent the average percent editing across individual edited regions altered in *tdp-1(ok803)* (73 altered/154 analyzed). Genomic location of edited regions, actual change in percent editing and actual *P*-values are listed in Supplementary Table S4.

B, C   Examples of hyper-edited intronic regions in *tdp-1(ok803)* mutant RNA-seq: intron 3 of ABC-transporter *haf-6* (B) [ChrI: 1170900-1173858] and intron 9 of FOXO transcription factor *daf-16* (C) [ChrI: 10774096-10774765]. Purple bars indicate annotated structural elements containing editing. Regions were divided into 10 equal windows and mean fraction edited for each window is plotted (*y*-axis). Connected points indicate editing in the same structural element. Percent editing in wild-type (blue), *tdp-1(ok803)* (red) and *adr-2(gv42)* (green) is shown. Scale bars (upper left, black), 100 bp.

"selected" in the IP. We identified 2,875 transcripts enriched in the wild-type J2-IP, representing 27% of all genes expressed in the experiment (10,630). Most (95%) of the enriched RNAs originated from protein-coding genes. Both antisense-overlap genes and A-to-I edited genes were significantly enriched ($P < 1 \times 10^{-15}$ hgd) by J2-IP (Fig 4D). As expected, transcripts targeted by endogenous small interfering RNAs (siRNAs), which often trigger siRNA production due to the presences of double-stranded regions (Fischer, 2010), were also enriched by J2-IP ($P < 1 \times 10^{-143}$, hgd, endo-siRNA target genes taken from Warf *et al*, 2012). Representative examples of these classes of enriched transcripts are shown as well as an unenriched gene (Fig 4E–G). Transcripts with the structural characteristics mentioned represented 73% of all coding genes enriched in the J2-IP (Supplementary Table S6).

## TDP-1 functions to limit the structure or stability of dsRNA

To identity transcripts with more dsRNA structure/stability in *tdp-1 (ok803)* mutants, we used DESeq software to compare normalized (to input) expression levels of all genes expressed in the J2-IP between *tdp-1* mutants and wild-type (Supplementary Fig S10). After filtering out low expressed genes and genes not enriched by J2-IP over input in either sample (see Materials and Methods and Supplementary Methods), we identified 1,422 genes with increased J2-IP enrichment ($P < 0.05$, FDR < 0.1) in *tdp-1* mutants compared to wild-type, representing about 40% of all transcripts selected by J2-IP. In contrast, only 166 genes (5% of all J2-IP selected transcripts) were identified as having less dsRNA structure/stability in *tdp-1* mutants (Fig 5A; Supplementary Table S7). This result

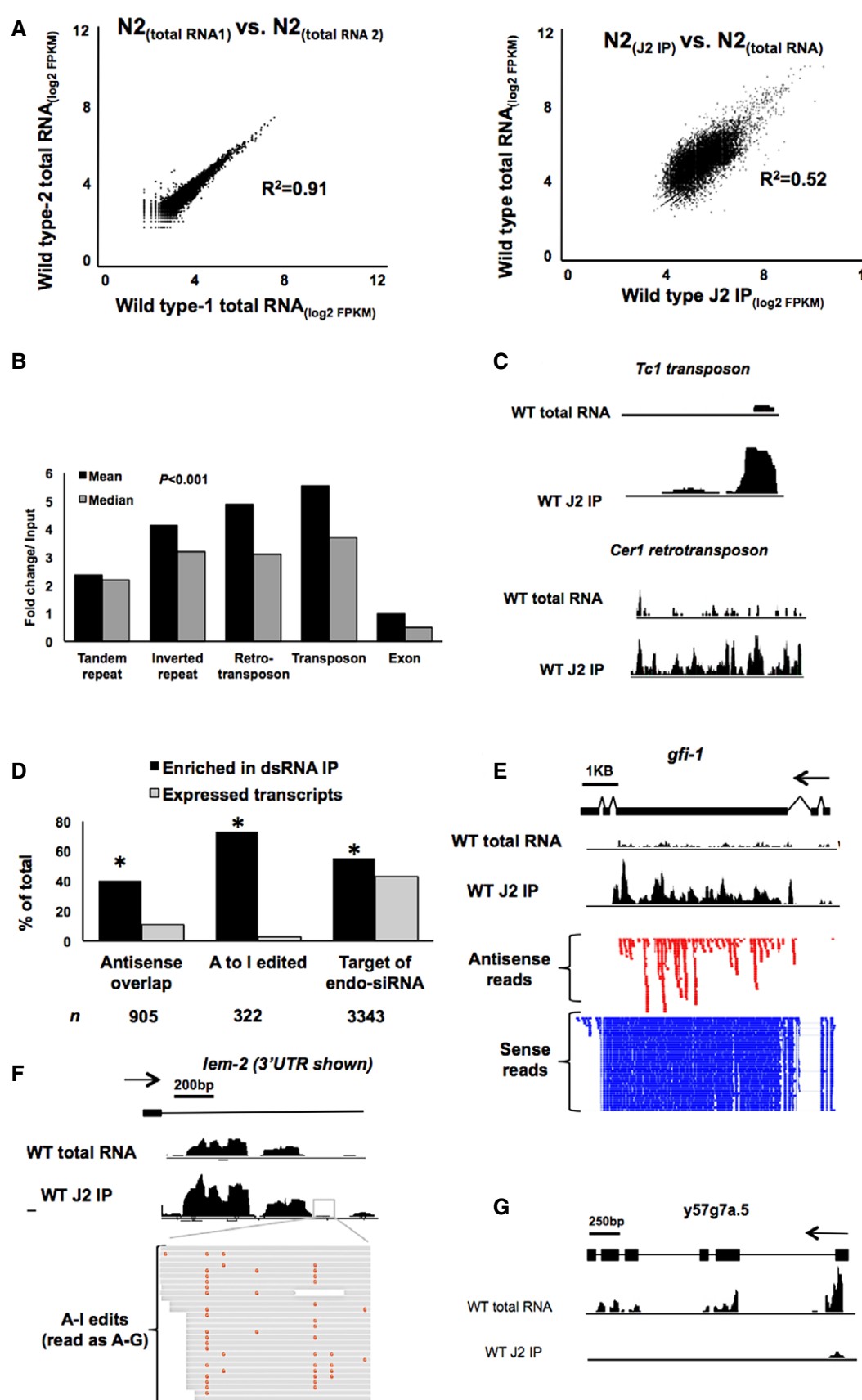

**Figure 4.**

**Figure 4. Immunoprecipitation with J2 antibody selects for dsRNA in wild-type worm extracts.**

A The J2 anti-dsRNA antibody immunoprecipitates a subset of RNAs from wild-type extracts. Plotted are RPKMs$_{(log2)}$ for two independent wild-type input RNA-seq datasets (left graph) and for a J2-IP RNA-seq dataset versus input RNA (right graph). 9,086 transcripts are shown. $R^2$ values for each plot are displayed in the graph.

B The fold change (in RPKM) over input (y-axis) of all significantly enriched repetitive transcripts (type indicated on x-axis) and all exons (corrected $P < 0.001$) in wild-type J2-IP RNA-seq. Graph shows both mean (black bars) and median (gray bars) fold enrichment/input.

C Coverage tracks (black bumps) of alignments to consensus sequences of representative transposons enriched in J2-IP RNA compared to total RNA. Coverage track height was set to the same value for both input RNA and J2-immunoprecipitated RNA, and height is proportional to abundance of each region.

D The percentage of total transcripts (gray bars) with potential dsRNA of three types (antisense overlap determined from Thierry-Mieg & Thierry-Mieg, 2006, A-to-I RNA editing taken from Supplementary Table S4 and siRNA targets taken from Warf *et al*, 2012) and the percentage of each of those transcript types enriched by J2-IP (in three independent biological replicates; black bars, *P-value $< 0.001$ (hypergeometric distribution); n = number of genes analyzed in each structural category).

E–G Examples of transcripts enriched in the J2-IP RNA-seq. The *gfi-1* locus (E) expresses an antisense transcript, as shown by both antisense (red) and sense (blue) reads. The 3′ UTR of *lem-2* (F) undergoes dsRNA-specific A-to-I RNA editing. The editing pattern in the boxed region is shown. (G) Example of transcript not enriched by J2-IP (*y57g7a.5*). Black arrows indicate direction of transcription.

indicates that TDP-1 limits the structure or stability of many dsRNA transcripts, resulting in accumulation of dsRNA when TDP-1 is absent. We confirmed the enrichment of several transcripts in the *tdp-1(ok803)* J2-IP by RT–PCR and showed this enrichment is abolished by pretreatment with dsRNA-specific RNAse (Supplementary Fig S9).

Analysis of transcripts increased in *tdp-1* mutant J2-IP did not indicate that any specific class of dsRNA (e.g. antisense to another gene or target of siRNA) was being affected in *tdp-1* mutants. Rather, we found that most transcripts that form dsRNA structure simply have more dsRNA structure/stability in the absence of TDP-1 (examples of enriched antisense-overlap genes shown in Supplementary Fig S11A). However, the exception to this observation was A-to-I RNA edited transcripts, which were significantly under represented ($P = 1 \times 10^{-5}$, hgd, $P = 9 \times 10^{-260}$, chi-square) among genes increased in *tdp-1(ok803)* J2-IP. This result suggests that hyperediting of these transcripts could counteract any potential increases in dsRNA structure in *tdp-1* mutants.

To determine whether *tdp-1* controls the structure of a specific functional class of transcripts, we performed a gene ontology analysis of genes whose transcripts have altered dsRNA structure/stability in *tdp-1* mutants. Among genes with increased recovery in *tdp-1(ok803)* J2-IP, we found an enrichment for genes functioning in nucleosome assembly (mostly replicative histone transcripts, 59/66 increased in dsRNA structure) as well as genes involved in translation and protein modification/turnover (Supplementary Table S8). Examination of enriched replicative histone transcripts in *tdp-1* mutants showed that both sense and antisense transcripts originating from histone genes were observed in *tdp-1(ok803)* and wild-type total RNA-seq data. However, J2-IP recovered the antisense histone-region transcripts only in *tdp-1(ok803)* (Supplementary Fig S11B), suggesting that these transcripts form intermolecular hybrids whose structure or stability is preserved in the absence of TDP-1 function. Interestingly, genes with decreased recovery in *tdp-1(ok803)* J2-IP were enriched for neuronal-specific genes, many of which are highly edited (Supplementary Table S8, bottom). Since ADAR enzymes are prominently expressed in the nervous system (Savva *et al*, 2012), one interesting possibility is that an initial increase in dsRNA structure in *tdp-1* mutants results in excessive editing that ultimately leads to decreased dsRNA structure in these transcripts.

A major fraction of transcripts with increased enrichment in *tdp-1* mutant J2-IP also mapped to repetitive elements. Comparison of normalized expression levels (RPKMs) of annotated repeat regions (taken from wormbase) indicated that most repeat regions showed increased dsRNA structure/stability in *tdp-1* mutants compared to wild-type (Fig 5B; Supplementary Fig S12), with the most dramatically affected class being tandem repeats. 60% of all expressed tandem repeats showed increased recovery in *tdp-1(ok803)* J2-IP compared to only 5% in wild-type. An example of a tandem repeat increased for dsRNA structure/stability is shown in Fig 5C.

## Transcripts with excess secondary structure are aberrantly represented in mature RNA

If excess dsRNA structure/stability results in altered levels of mature transcripts, we would predict transcripts that are over/under-expressed in the poly(A)-selected pool would be preferentially recovered by J2-IP. Indeed, transcripts aberrantly represented in poly(A) RNA-seq from *tdp-1(ok803)* animals were significantly enriched ($P = 1.7 \times 10^{-17}$, hgd) for transcripts isolated by J2-IP from *tdp-1(ok803)* (Supplementary Table S7 and Supplementary Fig S13). This result is consistent with the hypothesis that increased secondary structure within the pre-mRNA interferes with normal mRNA maturation causing altered representation in poly(A)-selected mRNA. As an example, Fig 5D depicts *col-119* (encodes a collagen protein), which is transcribed antisense to another expressed transcript. In the mature poly(A) RNA pool, this transcript is decreased. However, *col-119* transcript is overrepresented twofold in total RNA, suggesting that the reduction in poly(A) RNA results from defective maturation, as opposed to decreased expression/transcription. J2-IP enriched this transcript almost threefold in *tdp-1(ok803)* compared to wild-type controls, indicating this transcript has a high degree of aberrant dsRNA structure/stability that interferes with its maturation.

## TDP-1 functions to limit the double-stranded structure of intronic regions

TDP-43 homologs sustain the expression of transcripts with long introns presumably by binding the intronic sequences and allowing expression or stabilization of the transcript (Polymenidou *et al*, 2011). We propose that TDP-1 suppresses double-stranded structure within introns and may thereby prevent transcript degradation. To ask whether intronic sequences contain excess structure in *tdp-1* mutant animals, we compared normalized expression levels

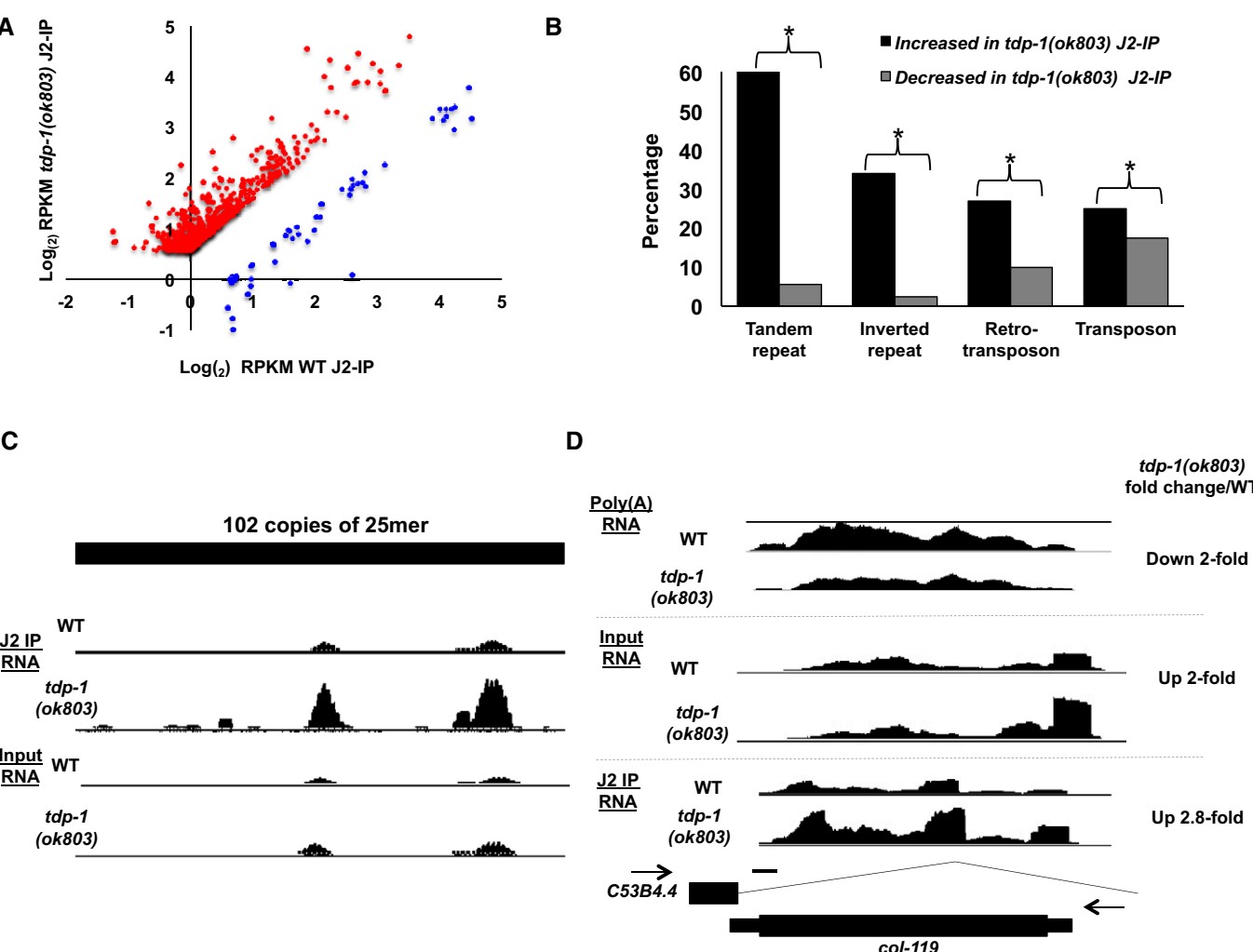

**Figure 5.　Double-stranded RNA transcripts are preferentially recovered in *tdp-1(ok803)* mutant extract.**

A　Graph comparing the $\log_2$ RPKM (normalized to input) of *tdp-1(ok803)* J2-IP (*y*-axis) versus wild-type J2-IP (*x*-axis) for all genes significantly increased (red dots) and decreased (blue dots) in representation ($P < 0.05$, FDR $< 0.1$ for all changes). Plot derived from data shown in Supplementary Fig S10 using three independent biological replicates of wild-type and *tdp-1(ok803)* J2-IP.

B　The percentage (*x*-axis) of all expressed repetitive elements (type shown on *y*-axis) that are significantly increased (black bars) and decreased (gray bars) in *tdp-1* mutant J2-IP compared to wild-type J2-IP ($P < 0.05$ for all changes). $*P < 1 \times 10^{-10}$ chi-square test. Plot derived from data shown in Supplementary Fig S12.

C　A representative example of a tandem repeat region that is increased for dsRNA structure/stability in *tdp-1(ok803)* J2-IP. Coverage tracks from both the J2-IP RNA-seq and input RNA-seq are shown; region displayed: chromosome I: 10,130,500–10,133,500.

D　Coverage tracks of *col-119* gene (transcribed antisense with the intron of expressed gene *C53B4.4*) in poly(A) RNA-seq (top), input (total) RNA-seq (middle) and J2-IP RNA-seq (bottom) (note: scale of coverage tracks for J2-IP is normalized to input for display purposes). Gene models for both genes are shown below coverage tracks, the direction of transcription is indicated by black arrows). Scale bars (black line), 200 bp. See also Supplementary Fig S11.

(RPKMs) for all intronic regions ($> 250$ bp) represented in the J2-IP RNA-seq. Among enriched regions, 1,287 introns were significantly different ($P < 0.05$, FDR $< 0.1$) between *tdp-1(ok803)* and wild-type, with 75% showing an increased abundance in *tdp-1(ok803)* J2-IP (Fig 6A; Supplementary Fig S14). Visual inspection of introns altered for J2-IP enrichment in *tdp-1(ok803)* mutants indicated that ~31% contained anecdotal evidence of A-to-I editing (scored by visual inspection of sequencing reads) and 80% overlapped with annotated complex and/or simple structural elements (see Supplementary Table S9). Both of these features are consistent with the formation of dsRNA. Many edited intronic regions observed in the J2-IP RNA-seq were not identified as edited in our initial analysis

(due to low coverage of intronic regions), suggesting that immunoprecipitation with the J2 antibody is capable of enriching for low abundance or rapidly degraded regions. Interestingly, introns with decreased recovery in *tdp-1(ok803)* J2-IP were highly enriched for A-to-I editing (~50%, $P = 4 \times 10^{-56}$, chi-square test) and conversely, regions with increased structure were unenriched for editing (~25%, $P = 5.8 \times 10^{-10}$, chi-square test), suggesting that hyper-editing in *tdp-1* mutants may result in a decrease in dsRNA in some intronic regions.

Double-stranded RNA structure, as well as A-to-I RNA editing, is known to modulate alternative splicing (Nishikura, 2010), so we asked whether transcripts with altered intronic dsRNA in the mutant

were the same transcripts with altered splicing in poly(A) RNA. Comparison of J2-IP enriched introns and genes with altered splicing in *tdp-1(ok803)* mutants revealed a significant enrichment (94/163 that could be assayed, $P = 7.5 \times 10^{-14}$, hgd) for intronic regions with differential recovery in *tdp-1(ok803)* J2-IP among transcripts altered in splicing. As a representative example, Fig 6B shows the *pqn-41* transcript, which displays enhanced inclusion of exon 18 in *tdp-1(ok803)* poly(A) RNA-seq and enrichment by J2-IP of the intronic regions bracketing that exon in *tdp-1(ok803)* mutants. A-to-I RNA editing was also observed in these introns consistent with the formation of dsRNA. Additional examples are provided (Supplementary Fig S15). These results suggest that some of the *tdp-1(ok803)* splicing abnormalities could be due to excess dsRNA formation and/or A-to-I editing within the intronic regions of transcripts altered in splicing.

## TDP-1 associates co-transcriptionally with genes whose dsRNA stability is increased in *tdp-1* mutants

To determine whether TDP-1 acts directly on transcripts with potential double-stranded structure, we asked whether TDP-1 protein associates with genes containing excessive dsRNA in *tdp-1(ok803)* mutants. As TDP-1 orthologs bind primarily intronic regions, we suspected that TDP-1 might associate with nascent pre-mRNAs co-transcriptionally in order to limit dsRNA structure or stability in these regions. Chromatin immunoprecipitation (ChIP) sequencing is widely used to determine the association site of RNA-binding/ processing factors that co-transcriptionally process nascent RNA (Swinburne *et al*, 2006). Therefore, we chose to analyze TDP-1 binding via chromatin immunoprecipitation followed by deep sequencing. In parallel, we also performed TDP-1 ChIP-seq on extract pretreated with RNase to confirm the ChIP signal was dependent on association with nascent RNA and not an association with the DNA, as has been done previously (Bieberstein *et al*, 2014). TDP-1 ChIP-seq was normalized using the RNase-treated ChIP-seq. Approximately 3,500 genes contained significant ($P < 0.01$) TDP-1 binding sites (Supplementary Table S10). We were not able to find any significant TDP-1 peaks not dependent on RNA (RNase ChIP sample was normalized using total genomic sequencing, data not shown) verifying that TDP-1 ChIP peaks represent co-transcriptional association of TDP-1 with nascent RNA. 64% of TDP-1 peaks were contained in introns that were 32-fold longer than the average intron length (mean size of bound intron: 2,120 bp versus 65 bp genome average). As TDP-43 orthologs show a preferential association with UG-rich sequences (Ayala *et al*, 2005), we asked whether worm TDP-1 also associates with UG repeats. Analysis of UG repeat sequences indicated TDP-1 associated with ~70% of all UG repeats of at least six repeats long ($n = 201$). Further, TDP-1 association increased with increasing UG-repeat size (Supplementary Fig S16). While this result indicates TDP-1 association is enriched in UG-rich regions, many TDP-1 bound regions did not contain UG repeats and not all UG repeats were bound by TDP-1, mirroring results reported for mammalian TDP-43 binding.

Analysis of TDP-1 binding in genes with increased dsRNA structure/stability in *tdp-1* mutants (as assayed by J2-IP enrichment) showed about 40% of these genes contained clear TDP-1 binding sites ($P = 1 \times 10^{-21}$, hgd). For J2-IP enriched intronic regions, TDP-1 binding showed a more significant enrichment (51% bound,

$P = 8 \times 10^{-141}$). A large fraction (37%) of TDP-1 ChIP peaks also overlapped with annotated repeat regions such as transposons and tandem repeats (2,061 peaks overlapped out of 5,587 peaks total), confirming that TDP-1 associates with potentially double-stranded transcripts and mirroring mammalian studies showing TDP-43 binds transposable elements (Li *et al*, 2012). An example of TDP-1 binding sites within intronic regions containing excess dsRNA structure/ stability is shown along with the J2-IP enrichment of these regions (Fig 6C). In order to provide independent support for our ChIP-seq results, we performed an RNA-immunoprecipitation (UV-CLIP) experiment using an antibody against TDP-1 followed by RT–PCR. As shown in Supplementary Fig S15, the RIP experiments confirmed TDP-1's association with a subset of introns that were bound by TDP-1 in the ChIP-seq. Taken together, these results support the model that TDP-1 associates co-transcriptionally with nascent pre-mRNAs to limit potential dsRNA structure, thereby allowing correct maturation of transcripts.

## TDP-1 maintains normal olfactory response by limiting the action of pathways that act on dsRNA

Our results indicated that TDP-1 functions to control the structure or stability of double-stranded RNA, but it is not clear whether there is a neurological consequence of excess dsRNA. Although *tdp-1* loss of function mutants are reported to have reduced mobility (Zhang *et al*, 2012), we were unable to detect movement deficits in *tdp-1(ok803)* animals under standard conditions (Supplementary Fig S17A and B). However, TDP-1 might function in other neuronally mediated processes, such as chemotaxis. We assayed the response of *tdp-1* mutant animals to attractive volatile compounds (butanone and isoamyl alcohol) that are sensed by the AWCR and AWCL neurons (Bargmann, 2006). Determination of chemotaxis indices (CI) revealed significant ($P < 0.001$) olfactory deficits in *tdp-1(ok803)* mutant animals (Fig 7A). Similar defects were also observed in the non-null allele, *tdp-1(ok781)* (Supplementary Fig S17C).

We were curious if the chemotaxis defects in *tdp-1* mutants were due to increased dsRNA accumulation or an alternate function of *tdp-1*. If aberrant dsRNA results in chemotaxis defects, it could either be because dsRNA itself is intrinsically deleterious or because the dsRNA leads to the hyper-activation of downstream processes, such as inappropriate production of endogenous siRNAs. To ask whether the *tdp-1(ok803)* olfactory defects were due to the RNAi pathway, we created *tdp-1(ok803)* strains that also had a loss of function mutation for a gene essential to the RNAi pathway, *rde-1 (ne219)*. Chemotaxis assays on this strain showed that mutation of *rde-1* completely suppressed the olfactory defect in *tdp-1(ok803)* animals (Fig 7B). Staining with the J2 antibody detected similar amounts of dsRNA in the *tdp-1(ok803), rde-1(ne219)* double-mutant as compared to the *tdp-1(ok803)* single-mutant worms (Fig 7C), confirming that the mutation in *rde-1* did not suppress the primary accumulation of dsRNA. These results indicate that specific neuronal defects in *tdp-1* mutants require the action of pathways downstream of dsRNA. Importantly, deletion of worm *adr-2* also results in a chemotaxis defect that can be rescued by mutation in *rde-1* (Tonkin & Bass, 2003), indicating that defective chemotaxis is a major phenotypic consequence of excess dsRNA in *C. elegans* and this defect is due to metabolism of double-stranded transcripts by the RNAi pathway.

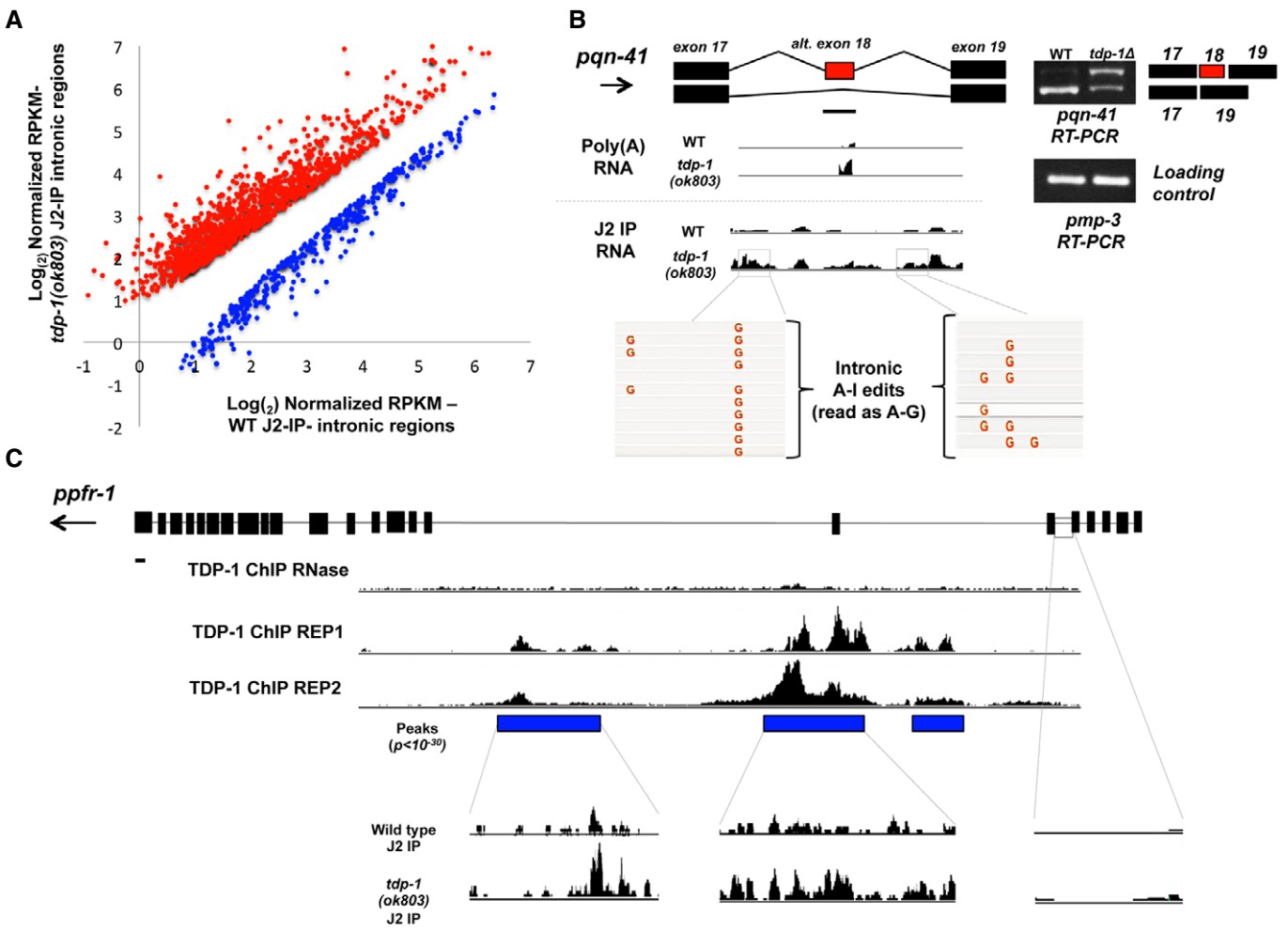

**Figure 6.  TDP-1 directly limits the dsRNA structure or stability of intronic RNA.**

A   Log₂ fold change of *tdp-1(ok803)* J2-IP/wild-type J2-IP (*x*-axis) for all expressed intronic regions selected in the J2-IP graphed according to increasing number of reads that map to each region (*x*-axis). Abundance levels for introns were normalized to input (see Materials and Methods for details). All introns selected in the J2-IP that are significantly ($P < 0.05$, FDR $< 0.1$, three biologically independent replicates) increased (red dots) and decreased (blue dots) between *tdp-1* mutant and wild-type J2-IP are shown. See Supplementary Table S9.

B   Example of a *tdp-1(ok803)* J2-IP enriched intronic region that exhibits altered splicing in *tdp-1(ok803)* polyA RNA-seq. Coverage tracks from poly(A) RNA-seq and J2-IP RNA-seq are displayed for *pqn-41* (ChrIII: 1,931,953–1,934,515). A-to-I RNA editing patterns in the boxed regions for each transcript are also depicted. Arrows depict the direction of transcription, and scale bars (solid black line below gene model) are set to 250 bp.

C   ChIP-seq pattern for TDP-1 within the long, structured gene *ppfr-1* (ChrI: 9,364,769-9,393,087). The location of significant TDP-1 peaks in both replicates as well as in the RNase control is shown by blue boxes below the ChIP-seq coverage tracks. The enrichment by anti-dsRNA immunoprecipitation (J2-IP) within regions bound by TDP-1 is displayed below significant peaks. Both wild-type (top) and *tdp-1(ok803)* (bottom) J2-IP tracks are shown to indicate excess RNA structure when TDP-1 is absent. The J2-IP enrichment for an intron not bound by TDP-1 is also shown. Arrows depict the direction of transcription, and scale bars (solid black line below gene model) are set to 250 bp.

Source data are available online for this figure.

## Limiting dsRNA accumulation is a conserved function of human TDP-43

TDP-1 limited the accumulation of double-stranded RNA in worms. To determine whether this is a conserved function of mammalian TDP-43, we treated human-derived M17 neuroblastoma or HeLa cells with antisense TDP-43 siRNA and then immunostained the cells with J2 antibody. Consistent with the worm results, we observed a dramatic increase in dsRNA accumulation in both cell types. Within HeLa cells (Fig 8A), the J2-immunopositive signal was concentrated in defined nuclear foci that were significantly

brighter ($P < 0.001$) in TDP-43 siRNA-treated cells than in control cells (Fig 8B). M17 neuronal cells also displayed increased J2 staining upon TDP-43 knockdown (Fig 8C), but the staining of dsRNA was more diffuse in these cells and occurred in both the nucleus and the cytoplasm. Quantification of immunofluorescence indicated that dsRNA was also significantly increased in M17 cells upon TDP-43 knockdown (Fig 8D). J2 staining specificity was confirmed by differential nuclease digestion (Supplementary Fig S18). The different subcellular localization of accumulating dsRNA in HeLa cells and M17 neuroblastoma cells suggests that TDP-43 could have distinct functions on dsRNA in these two cell types. Alternatively, different

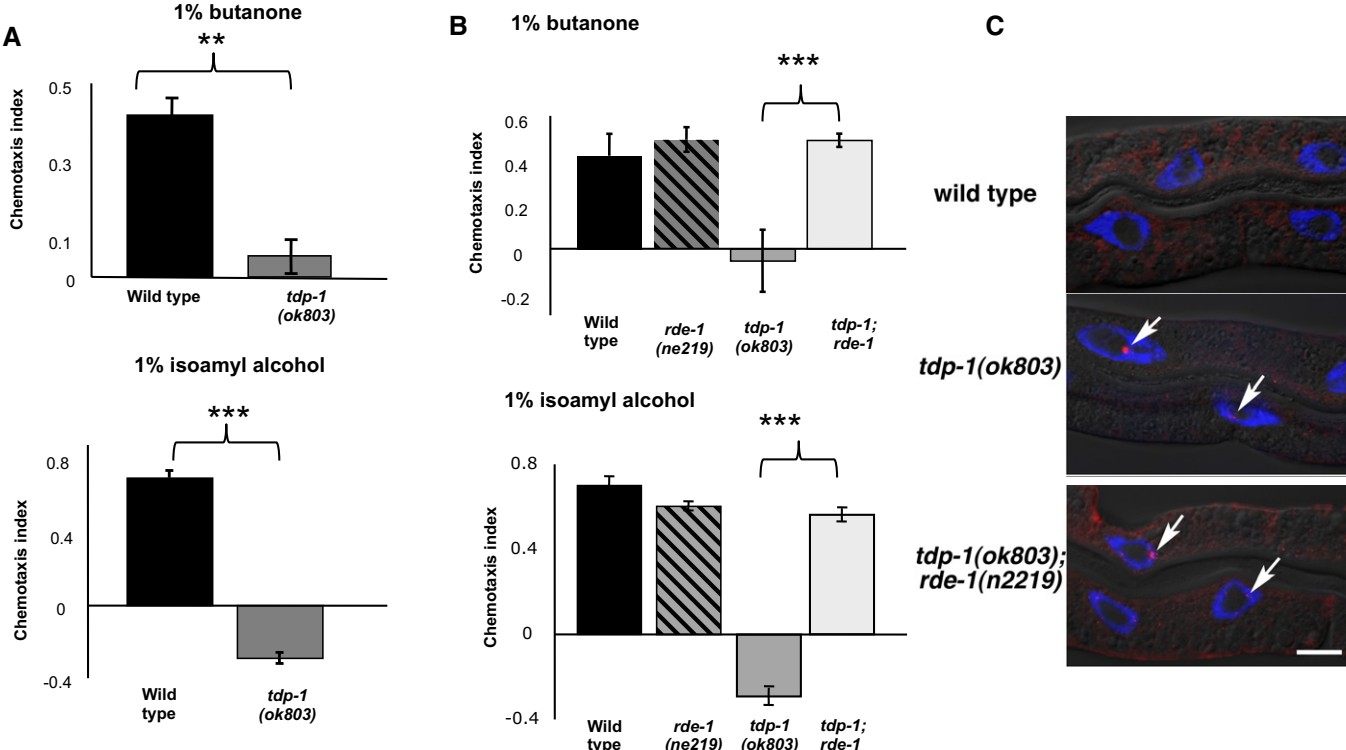

**Figure 7.  TDP-1 maintains chemotaxis by limiting RNA interference.**

A  The chemotaxis index (*y*-axis) for wild-type and *tdp-1(ok803)* animals toward 1% butanone and 1% isoamyl alcohol are shown. Negative values signify repulsion. Error bars represent SEM. **$P < 0.01$, ***$P < 0.001$ (Student's *t*-test).

B  Rescue of chemotaxis toward 1% butanone (top two graphs) and 1% isoamyl alcohol (bottom two graphs) in *tdp-1(ok803);rde-1(ne219)* double-mutant animals. Assays were done in triplicates at least 3 independent times. Error bars represent SEM. ***$P < 0.001$ (Student's *t*-test).

C  Immunostaining of isolated worm nuclei in wild-type, *tdp-1(ok803)* mutants and *tdp-1(ok803); rde-1(n2219)* double mutants. Single plane images distally overlaid with DIC image. The presence of J2 foci in *tdp-1(ok803); rde-1(n2219)* mutants indicates that deletion of *rde-1* does not suppress the accumulation of dsRNA in *tdp-1* mutants.

cell types may have distinctive mechanisms to metabolize or sequester dsRNA.

Increased dsRNA accumulation in TDP-43 knockdown cells suggests that human TDP-43 may also control dsRNA structure in the same transcript types observed in the *tdp-1* deletion. We do not predict that human TDP-43 will affect orthologs of worm transcripts as the common feature among affected transcripts is the inherent structure of the transcript (i.e., inverted repeats) not necessarily the function of its resultant protein. To ask whether TDP-43 knockdown in human cells resulted in increased dsRNA in the same transcript types as worm *tdp-1* deletion, we performed J2-IPs followed by RT–PCR of RNA isolated from TDP-43 knockdown cells and mock-treated controls. As our analysis in worms indicated that repetitive sequences and intronic regions were the most affected by *tdp-1* loss of function, we focused on these transcript types. As a representative example of a repetitive sequence, we assayed the Herv-K retrotransposon, which is misregulated in ALS pathological samples (Douville *et al*, 2011). Comparison of the intensity of the RT–PCR product for mock-treated cells and TDP-43 siRNA-treated cells indicated that TDP-43 knockdown dramatically increased the dsRNA structure of the Herv-K transcript (Fig 8E). We then asked whether intronic regions associated with human TDP-43 had increased J2-IP recovery in knock down cells by assaying Fus/TLS intron 8, which

is bound by TDP-43 in three locations (Tollervey *et al*, 2011). Again, the J2-IP signal in TDP-43 knock down cells was more pronounced than in mock-treated samples. Finally, we asked whether transcripts dependent on TDP-43 for correct splicing contained increased dsRNA within intronic regions. We selected five transcripts with reproducible alternative splicing differences in TDP-43 pathological tissue compared to healthy tissue (Tollervey *et al*, 2011). Of these five candidates, four showed increased recovery in the J2-IP from TDP-43 knockdown cells compared to control (Fig 8E; Supplementary Fig S19A). To confirm these results, we created RNA-seq libraries from an independent J2-IP on TDP-43 knockdown and mock-treated cells (along with input controls). After mapping and normalization to input control levels, we examined the J2-IP enrichment of regions tested by RT–PCR (Fig 8F; Supplementary Fig S19B). In all cases, we were able to confirm increased J2-IP enrichment in TDP-43 knock down (over input expression levels) compared to control. These results indicate that mammalian TDP-43 limits dsRNA structure in some of the same transcript types as worm TDP-1.

To rule out the possibility that dsRNA accumulation in human cells results from decreased ADAR2 function, we asked whether knocking down TDP-43 in human M17 neuroblastoma cells led to reduced GluR2 Q607R editing, an ADAR2-specific editing site

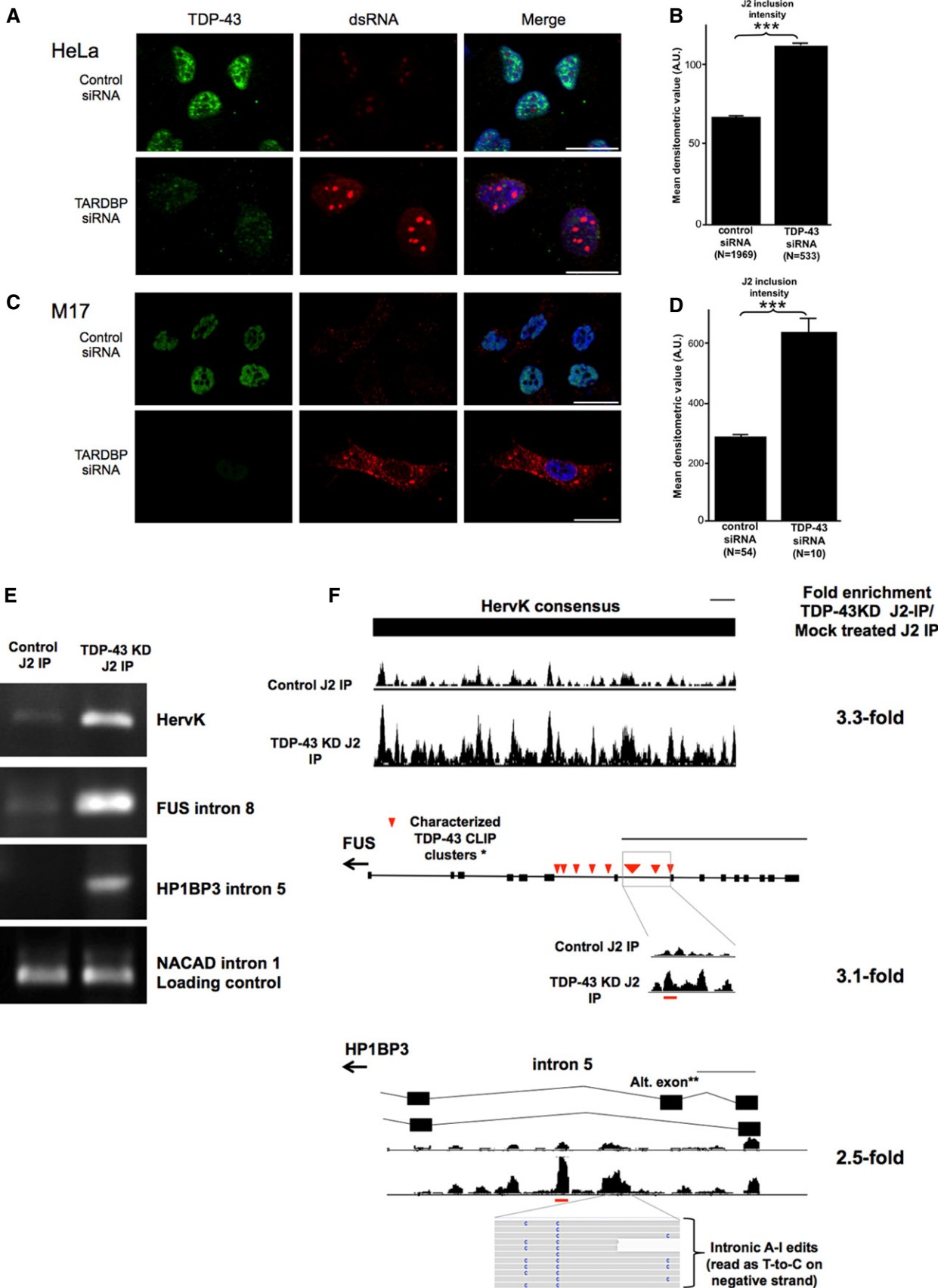

**Figure 8.**

**Figure 8.   Mammalian TDP-43 also functions to limit the accumulation of dsRNA.**

A   Cultured HeLa cells depleted for TDP-43 (green) with *TARDBP* siRNA display increased intensity of J2 anti-dsRNA labeling (red) within nuclear foci. Scale bars, 20 μm.

B   Densitometric quantification of the intensity of nuclear dsRNA foci. Graph shows the mean J2-dsRNA densitometric value, error bars = SEM. \***$P < 0.0001$ (Student's *t*-test). N = number of objects detected in each group.

C   Cultured M17 cells display increased nuclear and cytoplasmic dsRNA staining (red) upon knockdown of TDP-43. Scale bars, 20 μm.

D   Quantification of nuclear dsRNA in TDP-43 knockdown M17 cells. Graph shows the mean J2-dsRNA densitometric value, error bars = SEM. \***$P < 0.0001$ (Student's *t*-test). N = number of objects detected in each group.

E   Semi-quantitative RT–PCR of RNA precipitated from J2-IP on lysate from mock-treated M17 cells (left panel) and TDP-43 knockdown M17 cells (right panel). RT–PCR was done in triplicates (only one replicate shown).

F   Expression tracks from J2-IP RNA-seq in regions shown to have increased recovery in TDP-43 knockdown J2-IP compared to mock-treated controls. The gene model for each region is shown above the expression tracks. Evidence of A-to-I RNA editing in human sequencing read is also shown for HP1BP3. Herv-K expression tracks represent reads mapping to a consensus sequence of the Herv-K genome. All expression tracks are normalized to the number of reads in each sample, and the height of the track indicates reads depth. The fold enrichment by J2-IP in TDP-43 knockdown over control (normalized to total RNA-seq) for each region is shown to the right. The thin black line indicates scale, and scale is set to 500 bp. Red lines show the location of primer sequences relative to the gene. Expression tracks for NCAD1 (loading control) are shown in Supplementary Fig S20.

Source data are available online for this figure.

(Nishikura, 2010). We analyzed the editing by gene-specific RT–PCR followed by Sanger sequencing. We found no correlation between the amount of TDP-43 expression in M17 cells and the efficiency of editing in the GluR2 transcript (Supplementary Fig S20), indicating that dsRNA accumulation is not due to decreased ADAR2 function.

### TDP-43 is an RNA chaperone with strand displacement activity

Our data indicated that TDP-43 does not limit double-stranded RNA accumulation by maintaining A-to-I RNA editing or RNA interference, as both of these processes appear hyper-active in *tdp-1* deletion animals. Therefore, we postulated that TDP-43 may act directly on the RNA molecule as an RNA chaperone to limit dsRNA structure. In order to test this theory, we assayed recombinant human TDP-43 *in vitro* for strand displacement activity, an activity well characterized for many RNA chaperones, including human La protein (Naeeni *et al*, 2012). In brief, two complementary RNA oligos labeled with Cy5 or Cy3, respectively, were allowed to anneal at 37°C. Following annealing, an excess of unlabeled bottom strand was added, Cy3 was excited and the relative change in FRET (Cy5 excitation) was measured over time (see Fig 9A). In the presence of our negative control (BSA), the relative FRET signal increased (Fig 9B), which is expected as addition of competitor oligo results in an increase in the concentration of one of the partners in the annealing reaction (Rajkowitsch & Schroeder, 2007). However, in the presence of our positive control (human La), the relative FRET signal decreased over time due to the unlabeled bottom strand displacing the labeled strand, an activity that requires an RNA chaperone. Importantly, in the presence of recombinant human TDP-43, the relative FRET signal also significantly decreased ($P = 5 \times 10^{-31}$, paired *t*-test, BSA compared to TDP-43) at a level similar or greater than the decrease observed for human La (Fig 9B). This result indicates that human TDP-43 is capable of RNA chaperone activity *in vitro* and suggests that TDP-43 could function as an RNA chaperone on nascent RNA *in vivo* in order to limit inappropriate dsRNA structure/folding during transcription.

## Discussion

TDP-43 is involved in many different RNA processes, making the identification of this protein's role in neurodegeneration complex.

Because a large number of binding partners have been identified for TDP-43 (Freibaum *et al*, 2010), it is possible that TDP-43 affects many aspects of RNA metabolism by participating in multiple protein complexes. An alternative, although not mutually exclusive, possibility is that TDP-43 interacts with RNA in a way that is important for many different RNA-mediated processes. In this study, we uncovered a fundamental role for TDP-1 in directly controlling dsRNA accumulation and processing. Furthermore, we showed that knockdown of human TDP-43 also increases dsRNA accumulation, implying that functions described for worm TDP-1 have relevance to mammalian TDP-43.

The hypothesis that a fundamental function of TDP-1/TDP-43 is to control the accumulation of dsRNA is strongly supported by the studies of Polymenidou *et al*, who examined gene expression in mouse brain after TDP-43 knockdown. Strikingly, 29 of the top 50 transcripts overexpressed after TDP-43 knockdown are type 1 interferon-inducible genes (Samarajiwa *et al*, 2009) (Supplementary Table S12). Included in the up-regulated genes in TDP-43 knockdown mice were the mouse orthologs of PKR (up 3.9-fold), RIG-1 (up 4.1-fold) and MDA5 (up 3.4-fold), which specifically respond to cytoplasmic dsRNA (Li *et al*, 2011). Because interferon activation is a major consequence of excessive dsRNA mammals, it would be interesting to determine whether increased interferon activation contributes to neuronal toxicity in mammals.

While worms lack an interferon response, our data indicate that processes downstream of dsRNA (i.e., RNAi) result in the neuronal phenotypes of *tdp-1* mutants. In addition, Vaccaro *et al* (2012) demonstrated that the *tdp-1(ok803)* allele renders worms hypersensitive to oxidative stress and interacts with the *C. elegans* insulin-like signaling (ILS) pathway. Interestingly, *daf-16* (the key transcription factor controlling ILS in *C. elegans*) is hyper-edited in *tdp-1(ok803)* (Fig 3C), suggesting increased dsRNA structure in *tdp-1(ok803)* worms could contribute to the altered stress sensitivity.

### Identification of dsRNA by immunoprecipitation and deep sequencing

In this study, we utilized anti-dsRNA immunoprecipitation experiments to globally identify transcripts containing double-stranded structure. Isolation and sequencing of dsRNA from lysates have the advantage of identifying transcripts likely to contain

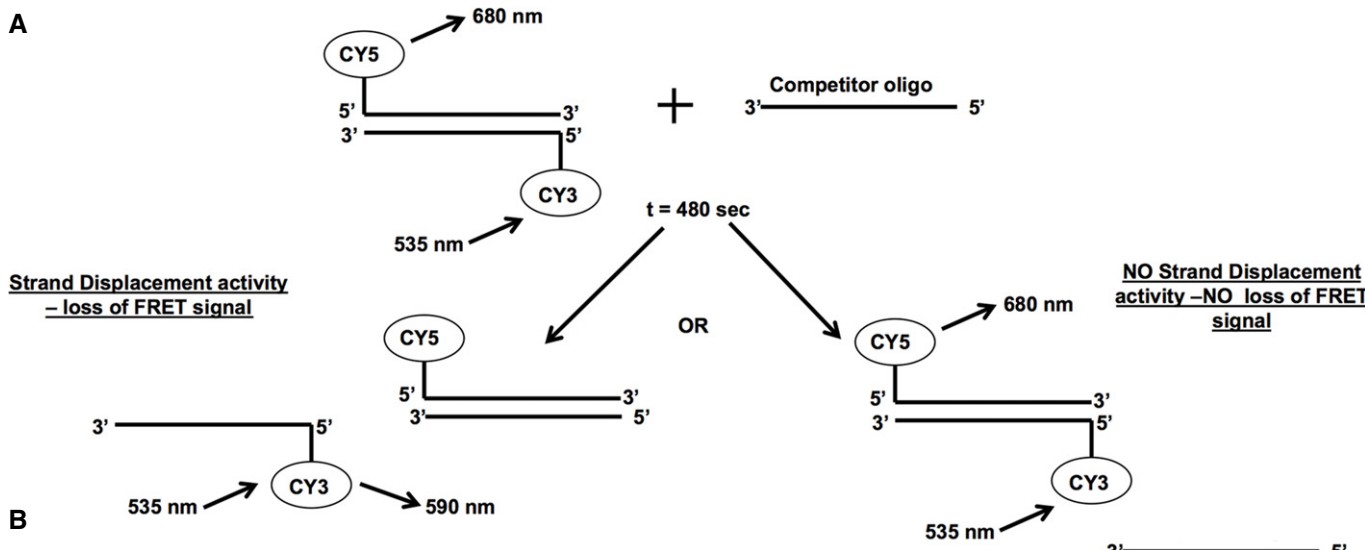

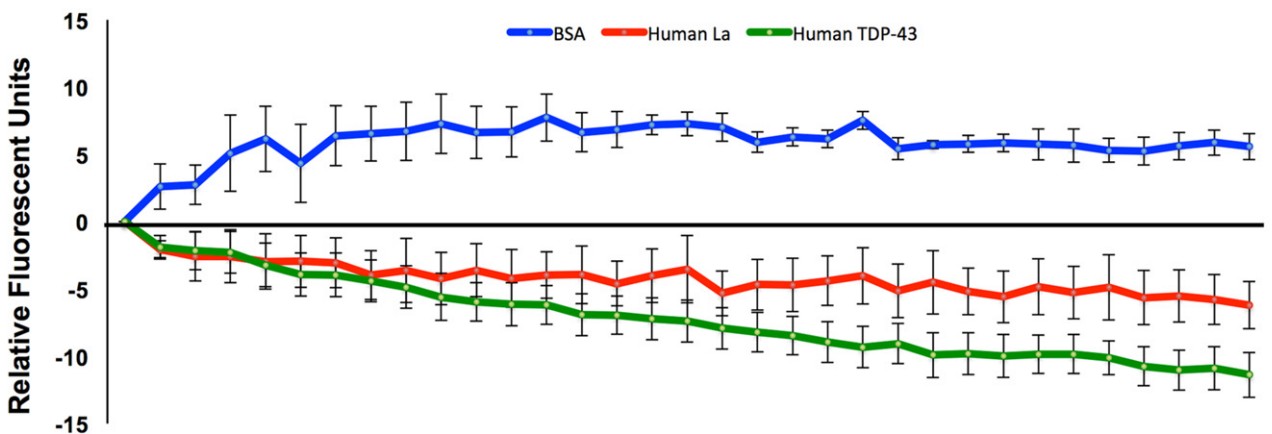

**Figure 9.  Human TDP-43 displays strand dissociation activity *in vitro*.**

A  Assay for strand displacement activity (adapted from Naeeni *et al*, 2012).

B  RNA strand dissociation activity was measured as a change in relative FRET signal (*y*-axis) over time (*x*-axis) of two complementary, fluorescently labeled (Cy5 and Cy3, respectively) RNA oligos (annealed for 480 s at 37°C) following the addition of tenfold molar excess of a competitor oligo complementary to the Cy5-labeled RNA molecule. RNA molecules were excited at 535 nm, and FRET signal was recorded at 680 nm. Initial FRET fluorescence was set to zero and then measured every 15 s for 480 s. Data represent results from six to seven independent assays. Error bars, SEM calculated between replicates at each 15 s time point. The assay was done in the presence of Bovine Serum Albumin (BSA) protein (blue line); recombinant human Lupus Antigen (Human La) (red line), and recombinant human Tar-DNA Binding protein 43 (TDP-43) (green line).

double-stranded RNA *in vivo*, in contrast to protocols that rely on isolation and refolding of RNA *in vitro* (Underwood *et al*, 2010). Refolding after RNA isolation could potentially disrupt RNA interactions or introduce structure that normally does not exist *in vivo*. While immunoprecipitation of dsRNA from lysates could recover transcripts that are not themselves double-stranded, but are associated with dsRNA, almost 75% of J2-immunoprecipitated transcripts had dsRNA characteristics. This result indicates that this method does preferentially identify transcripts containing dsRNA. In addition, anti-dsRNA immunoprecipitation isolated a significant amount of pre-mRNA or excised intronic sequences allowing the analysis of structured transcripts that affect pre-mRNA processing, but are normally too transient to detect.

**Characteristics of transcripts with increased double-stranded RNA structure in *tdp-1* mutants**

Immunoprecipitation of dsRNA in *tdp-1* mutants indicates that many transcripts rely on TDP-1 to limit intra-strand and/or inter-strand structure or stability. The variety among these transcripts is quite striking, such that dsRNA immunoprecipitation in *tdp-1* mutants enriched for both coding and intronic RNAs, as well as most replicative histone transcripts. We do not know which of these transcripts constitute the J2-immunopositive foci, or whether foci in different cell types contain different transcripts. The visible dsRNA foci in TDP-43 knockout/knockdown likely contain only a sub-population of accumulating dsRNA molecules that are retained

(or persist) due to innate characteristics of those transcripts. Importantly, expression of tandem array transgenes in *C. elegans*, which is known to produce dsRNA (Knight & Bass, 2002), can also lead to the accumulation of J2-immunoreactive nuclear foci (C. D. Link, unpublished results). This observation suggests both that J2 foci can result from accumulation of a limited set of transcripts, and the detection of J2-immunoreactive foci does not require removal of endogenous TDP-1 (as would be expected, for example, if TDP-1/TDP-43 binding to transcripts obscured epitopes recognized by the J2 antibody).

Importantly, the transcripts enriched by J2-IP in *tdp-1* mutants undergo distinctly different processing events. Some of the enriched transcripts undergo pre-mRNA splicing (most mRNAs), and some do not (replicative histone transcripts and ncRNAs). Similarly, ncRNAs and replicative histone transcripts have unique 3′-end formation pathways that differ from the canonical cleavage and polyadenylation machinery used by mRNAs (Wilusz & Spector, 2010). Given the varied maturation pathways of transcripts whose double-stranded structure is decreased by TDP-1, it is unlikely that TDP-1 reduces this structure by controlling steps in RNA processing (i.e., splicing or 3′-end formation). Rather, TDP-1 is likely to play a fundamental role in limiting RNA structure of many origins in a variety of transcript types.

### Transcript abundance and splicing in *tdp-1* mutants

In the absence of *tdp-1*, potentially structured transcripts are both increased and decreased in the mature (poly-A) RNA pool. These same transcripts are associated with TDP-1 and contain increased amounts of dsRNA structure/abundance in *tdp-1* mutants. We do not know the exact mechanism by which increased dsRNA formation/stability alters the abundance of potentially structured mRNAs in *tdp-1* mutants. However, increased structure could either increase or decrease transcript abundance depending on the individual regulatory pathways acting on each transcript. For example, increased structure could trigger siRNA production, which is known to down-regulate transcript abundance of antisense-overlap transcripts in multiple systems (Ghildiyal *et al*, 2008; Gu *et al*, 2009; Gullerova *et al*, 2011). Alternatively, increased intra- or inter-molecular structure in the RNA could preclude association of factors that promote degradation (such as micro-RNAs or RNA-binding proteins), increasing transcript abundance. Consistent with both of these possibilities, many aberrantly represented overlapping gene pairs in *tdp-1* mutants are co-altered, such that if one gene in the pair is increased/decreased, the other gene in the pair is also increased/decreased.

Further, inappropriate formation of dsRNA in introns could result in instability and degradation of those transcripts through the action of dsRNA-binding proteins and dsRNA nucleases. Nuclear dsRNA-binding proteins are known to destabilize structured pre-mRNA (Danin-Kreiselman *et al*, 2003; Kadener *et al*, 2009). Our data also indicate that a large number of excessively structured introns display altered splicing patterns in *tdp-1* mutants. The intronic A-to-I RNA editing could contribute to the splicing abnormalities as editing is known to alter splice site recognition (Laurencikiene *et al*, 2006). dsRNA structure itself also affects splicing through multiple mechanisms (Warf & Berglund, 2010). Formation of a dsRNA stem-loop is responsible for the skipping of exon 7 in the *Smn2* transcripts. Interestingly, TDP-43 binds *Smn2* pre-mRNA and stimulates inclusion of this same exon (Singh *et al*, 2007). Whether TDP-43 affects the structure of SMN2 RNA, however, is currently unknown.

### How does TDP-1 limit dsRNA accumulation?

Data presented here indicate that TDP-43 can act as an RNA chaperone *in vitro*. TDP-43 may function in this capacity to maintain correct RNA folding during active transcription. In the absence of TDP-43, inappropriate RNA folding could result in abberant dsRNA structure leading to defects in RNA processing and degradation. We cannot rule out the possibility that TDP-43 limits dsRNA accumulation through additional or alternative pathways. TDP-43 could also act as an RNA chaperone post-transcriptionally. TDP-43 has been shown to localize to a variety of RNA-containing cellular bodies in both the nucleus and cytoplasm (Da Cruz & Cleveland, 2011). Perhaps, TDP-43 maintains the function of these bodies by ensuring the correct structure of the transcripts contained within. Importantly, TDP-43 is a known component of nuclear paraspeckles, which are nuclear bodies that retain edited and structured RNAs (Fox & Lamond, 2010). TDP-43 could control the retention/release of transcripts in paraspeckles by altering RNA structure. Consistent with this possibility, mouse *Slc7a2* RNA, which was the first transcript identified as retained in paraspeckles due to an edited hairpin in the 3′ UTR, is bound by mouse TDP-43 in four distinct regions within the 3′ UTR (Polymenidou *et al*, 2011). Interestingly, FUS/TLS and Matrin3, other RNA-binding proteins causative in ALS (Kwiatkowski *et al*, 2009; Johnson *et al*, 2014), are also components of paraspeckles (Naganuma *et al*, 2012).

### Is dsRNA accumulation relevant to ALS/FTLD pathology?

TDP-43 is aberrantly localized to the cytoplasm in several pathological situations. Whether dsRNA is involved in this aberrant localization is an intriguing question. TDP-43 mislocalization is associated with CUG repeat expansion diseases (Schwab *et al*, 2008; Hart & Gitler, 2012), as well as with the recently identified *C90orf72* ALS-FTD mutation, which consists of an expanded GGGGCC repeat (DeJesus-Hernandez *et al*, 2011). Transcripts containing either of these RNA repeats form extensive dsRNA structures. Furthermore, age-associated decreases in the mouse ADAR2 ortholog is concurrent with aberrant TDP-43 localization in motor neurons (Hideyama *et al*, 2012). Decreased ADAR2 function would be predicted to increase nuclear dsRNA. Finally, *NEAT1* RNA, which nucleates paraspeckles, is induced in mouse brains by dsRNA viruses (Saha *et al*, 2006) and is highly up-regulated in human brains affected by FTLD (Tollervey *et al*, 2011), consistent with a potential increase of dsRNA accumulation in FTLD pathology. Therefore, future studies to determine whether aberrant dsRNA formation causes, or is caused by, the pathological mislocalization of TDP-43 are paramount.

## Materials and Methods

### *Caenorhabditis elegans* strains

Maintenance and growth of worms were performed as described in (Brenner, 1974), and all strains were raised at 20°C unless otherwise

noted. All transgenic strains used in this study were created by gonad injection and integration of DNA array. Supplementary Table S12 lists all strains created or used in this work.

## Chemotaxis assays

Chemotaxis assays were performed as described previously (Kauffman et al, 2011). 100-mm agar plates were spotted with 1 μl of attractant diluted 1:100 or 1 μl dilution buffer and 1 μl 1 mM sodium azide to paralyze worms arriving in either attractant or buffer-only spot. Worms were synchronized by alkaline hypochlorite treatment and hatched off overnight in S-basal buffer. Animals were then allowed to grow to 1-day-old adults at 20°C (about 72 h). Approximately 200 animals were added to each assay plate in a defined origin position in 50 μl of S-basal buffer, released simultaneously, and worms were allowed to crawl freely for 2 h. The total number of animals in the attractant spot, buffer-only spot and in undefined locations around the plate were counted. The chemotaxis assay was determined according to the equation: $(A_{(\# \text{ of worms in attractant spot})} - C_{(\# \text{ of worms in buffer only spot})})/$ total number of worms in the assay.

## Thrashing assays

Liquid thrashing assays were performed using synchronized 1-day-old adults grown at 20°C. Thrashes were counted for 30 s under a dissecting microscope by hand. For automated thrash counting, C. elegans body bends were captured using the wrMTrck plugin developed by Jesper S Pedersen (http://www.phage.dk/plugins/wrmtrck.html). Body bends of 4 L4 worms thrashing in 1× S basal were captured for 30 s. Three independent biological replicates were analyzed for wild-type and tdp-1(ok803) animals, resulting in a total of 12 worms per group. All the movies were filtered as suggested in the manual prior to body bend analysis.

## Caenorhabditis elegans immunoblotting and immunohistochemistry

Immunoblotting of worm and human proteins was done according to Ash et al, 2010. 20 μg of total protein was loaded per lane, and the resultant blots were probed with primary antibodies against anti-TDP-1 (1:2,000, in house), anti-tubulin (1:4,000, Sigma), anti-TDP-43 (1:2,000, Abnova) or anti-GAPDH (1:4,000, BioDesign). Blots were probed with secondary HRP-conjugated antibodies (1:5,000, Jackson) and developed in ECL Plus (Amersham). For immunohistochemistry, permeabilized whole-mount preparations were probed with J2 antibody (English & Scientific Consulting Lot: J2-1102 and J2-1103) at 2–5 μg/ml for 16 h at 4°C and Alexa dye-conjugated goat anti-mouse secondary antibody (Invitrogen) at 4–10 μg/ml for 2 h at room temperature. To assay the sensitivity of J2 immunoreactivity, permeabilized worms were incubated with either RNase V1 (0.003 U/μl, 30 min at 37°C) or RNase T1 (30 U/ml, 30 min, room temperature) in RNA structure buffer (all from Invitrogen, AM2275). RNase-treated worms were washed and probed with J2 as described above. Images were acquired with a Zeiss Axiophot microscope equipped with digital deconvolution optics (Intelligent Imaging Innovations), and image brightness and contrast were digitally adjusted in Photoshop.

## Extracts

Extracts for J2-IP experiments were made from young adult animals grown on solid plates at 15°C. Worms used for TDP-1 immunoprecipitation followed by RT–PCR were cross-linked according to Zisoulis et al, 2010. Worms used for J2-IP were not cross-linked. Extract was prepared by bead beating as described in Saldi et al, 2007. Extracts were used immediately for immunoprecipitations or dissolved in TRIzol (for input RNA).

## RNA immunoprecipitations

Immunoprecipitations were preformed as described (Saldi et al, 2007). 20 μl of Protein A magnetic beads (Dynabeads, Invitrogen) were washed, blocked, and bound by antibody at a ratio of 10 μl/IP for both anti-TDP-1 (made in house) and anti-J2. Protein concentration of each extract was determined by Bradford assay (Bio-Rad 500-0111), and 100 μg of protein was added to each IP. dsRNase-treated extract was incubated with 1 μl of RNase V1 for 30 min (room temperature) prior to addition to beads. Tubes were rocked at 15°C for 2 h (anti-J2 IPs) or 4°C overnight (anti-TDP-1 IPs). The supernatant was removed, beads were washed, and immunoprecipitated RNA was removed from beads by TRIzol extraction. DNA was removed from immunoprecipitated RNA using TURBO DNase (Invitrogen) and dissolved in nuclease-free water.

## Chromatin immunoprecipitation (ChIP)

ChIP was performed as described (Garrido-Lecca & Blumenthal, 2010). 20 μl TDP-1 antibody was used. DNA was purified using a QIAGEN column (QIAquick, # 28104) and eluted twice with 30 μl of water. For RNase-treated extract, a combination of RNases (RNase A, T1 and V1) was added to the extract and rocked for 15 min at room temperature prior to addition to the ChIP. Completion of RNase was confirmed on an aliquot of treated and untreated extract on acrylamide gel (data not shown).

## Cell culture and immunohistochemistry

M17 neuroblastoma and HeLa cells were cultured in OptiMEM (Invitrogen), and when confluent, $1.75 \times 10^4$ cells/well plated to 24-well plates on coverslips. TDP-43 was knocked down for 72 h with 20 nM TARDBP siRNA (QIAGEN) (Prudencio et al, 2012) or All Star control siRNA (QIAGEN) and siLentFect reagent (Bio-Rad) and fixed in 4% PFA in DEPC-treated PBS and permeabilized. Nuclease treatment of HeLa cells was performed at 37°C for 30 min with 100 U/ml of RNase-free DNase I and RNase A (QIAGEN), ShortCut RNase III (NEB) or ultrapure benzonase nuclease (Sigma) each in their recommended buffers. Protein Block Serum Free (Dako) was used for blocking. Primary antibody incubation was done overnight in Antibody Diluent (Dako) with 0.25 μl/ml RNase OUT and 1:1,000 anti-TDP-43-Cterm (Protein-Tech) and 1:1,000 J2, and secondary antibodies were 1:1,000 (donkey) anti-rabbit-AF488 and anti-mouse-AF568 (Alexafluor). Coverslips were counterstained with 0.1 μg/ml Hoechst. Images were taken by Zeiss AxioImager Z1 with Apotome at 63× with fixed exposures times.

### RNA isolation, cDNA library preparation, high-throughput sequencing

RNA for poly(A) and total RNA sequencing libraries was extracted from whole animals by TRIzol extraction. Genomic DNA was removed using TURBO DNase (Invitrogen). For total RNA libraries, 5 µg of RNA was run through a Ribo-Zero column (Epicenter, #R2C1046) to remove ribosomal RNA. For poly(A)-selected libraries, RNA was selected using sera-mag magnetic oligo dT beads (Therma Scientific). Libraries were created using Illumina TruSeq kits (RS-122-2001). RNA recovered by immunoprecipitation with the J2 antibody (three biologically independent lysates) of young adult worms as well as input material (as a loading control) was converted into strand-specific total RNA libraries using V2 Scriptseq (Epicenter #SSV21106) kits following manufacturer's instructions, except reverse transcription was done with Super-Script III (Invitrogen #18080-044) using incrementally increasing temperatures from 42 to 59°C to allow for transcription though structured RNAs. rRNA was not removed from J2-IP RNA samples. Immunoprecipitated DNA from ChIP samples was converted in sequencing libraries using ChIP-Seq DNA Sample Prep Kit (IP-102–1001). Libraries were sequenced on Illumina HiSeq 2000 platforms. Data were deposited under GEO accession number GSE61581.

### Sequencing alignment, gene expression quantification, and J2-IP analysis

Detailed algorithms and specifications for sequencing analysis including read filtering, mapping, and differential expression calculations are provided in the Supplementary Methods and Supplementary Fig S14. In brief, reads were trimmed and filtered, and rRNA reads were removed. Reads were aligned to the WS220 *C. elegans* genome. Differential expression in poly(A) selected libraries between wild-type and *tdp-1(ok803)* mutant was calculated using the Bioconductor package edgeR (release 2.12) (Robinson *et al*, 2010). Identification of transcripts enriched by J2-IP in wild-type animals was done by comparing RPKMs of expressed genes/repeat regions in the J2-IP RNA-seq to input RNA-seq. Genes/repeat regions that were significantly (corrected $P < 0.05$) increased in abundance > 1.5-fold over input were considered enriched by J2-IP. To determine genes, repeat regions and introns differentially selected by J2-IP in wild-type versus *tdp-1(ok803)* extract (three biological replicates of each), read counts for well-expressed regions were normalized to input (see Supplementary Methods) and read into DESeq v1.14.0. A significance cut-off of $P < 0.05$ and FDR < 0.1 was used.

### Identification of *adr-2*-dependent editing locations

For wild-type, *adr-2(gv42)*, and *tdp-1(ok803)* total RNA samples, the alignments were piled up using an in-house program. Positions in wild-type and *tdp-1(ok803)* samples with a discrepancy between the genomic sequence and mRNA sequence consistent with A-to-I editing (A-G and C-T transitions) were identified. Potential edited positions were compared to the *adr-2(gv42)* sample and sites also showing A-to-G or C-to-T changes were discarded. To identify editing in introns enriched by J2 immunoprecipitation (not identified as

edited in our initial analysis of total RNA due to insufficient read depth), reads mapping to all J2-IP enriched introns were inspected visually on the IGV genome browser and hand-annotated as edited if editing was well represented and was present in all three biological replicates. Additional details in Supplementary Methods.

### Calculation of RNA editing frequency in *tdp-1(ok803)* mutants compared to wild-type

The Integrative Genome Viewer (IGV) tool, count, was used to transform alignment files into pileups for downstream analysis. Well-expressed (> 20 reads) edited regions were used to compare percent editing between wild-type and *tdp-1(ok803)*. The total number of edited nucleotides was divided by the total number of potentially edited nucleotides (determined by evidence of editing in either sample) for each location. Significant changes were determined by performing a two-proportion *z*-test between wild-type and *tdp-1(ok803)* percent edited ratios; $P < 0.05$ for significant changes.

### Gene ontology term enrichment

David (Huang da *et al*, 2009) was used to calculate enriched gene ontology (GO) terms. A list of unranked significantly up- and down-regulated genes was compared to a background list containing all *C. elegans* genes expressed in our RNA-seq samples.

### Splicing analysis

To capture splicing changes between wild-type and *tdp-1(ok803)* mutants, poly(A)-selected RNA from each strain was resequenced 100 bp paired-end at a high read depth (~280 million reads per sample) to achieve the depth and mapping specificity to analyze alternative splicing. The Cuffdiff (v1.30) differential splicing test was used for splicing analysis.

### ChIP peak identification

Filtered and trimmed ChIP-seq samples were aligned to WS220 using Bowtie. Significant TDP-1 ChIP peaks were identified using MACS software (version 1.42) (Zhang *et al*, 2008). Two replicates of TDP-1 ChIP-seq were done, and a TDP-1 ChIP-seq pretreated with RNase was used as the control file. Bandwidth was set to 300, a 2-fold to 30-fold change over control was required, and the *P*-value cut-off was set at $P < 0.01$. Regions positive for TDP-1 peaks were counted only if an overlapping peak existed in both biological replicates, and there was no peak in the RNase-treated sample.

### Strand displacement assay

Strand displacement assay was done as described in Rajkowitsch & Schroeder (2007) with the following modifications: Labeled and competitor RNA oligos were obtained from IDT, and recombinant TDP-43 and La protein were obtained from GenWay Biotech, Inc. RNA oligo sequences were taken from Naeeni *et al*, 2012. Initial annealing was done at 37°C for 480 s in 40 µl assay buffer. Labeled oligos were used at a final concentration of 1 µM, and recombinant protein was used at 10 µM. Competitor oligo was added in 10 µl for

a final concentration of 100 μM in 50 μl. The mixture was shaken for 2 s, excited at 535 nm, and florescence was read at 690 nm every 15 s for 480 s at 37°C. The assay was done in a 384-well black microtiter plate (Greiner Bio-one). All measurements were taken in a SpectraMax M5 microplate reader machine, and SoftMax Pro software was used to calculate relative florescent units for each sample. The assay was done seven independent times, and replicates deviating more than two standard deviations from the mean were removed, such that data represent six replicates of BSA, six replicates of human La and seven replicates of human TDP-43.

**Supplementary information** for this article is available online: http://emboj.embopress.org

## Acknowledgements

We would like to thank Kyle Dack for assistance with the chemotaxis assays. Some nematode strains were provided by the Caenorhabditis Genetics Center, funded by the NIH National Center for Research Resources. This work was supported by NIH grants NS063964 (CDL), GM42432 (TB), AG026251, NS063964, ES20395 (LP), DOD grants W81XWH-10-1-0512-1 and W81XWH-09-1-0315AL093108 (LP), and the CIHR Frederick Banting and Charles Best Canada Graduate Scholarship (GW).

## Author contributions

TS and CL conceived the project, designed the experiments, and wrote the manuscript. TS produced data in Figs 1 and 3–9 and most Supplementary Figures and contributed to bioinformatics analysis. CL produced data in Figs 2 and 7, Supplementary Figs S4 and S5. PA produced data in Fig 8, Supplementary Figs S18 and S20. GW and LS contributed to data in Fig 3. PG did the bulk of the data processing and bioinformatics. AG contributed to data in Fig 6. CR produced data in Supplementary Figs S1 and S2. VD contributed to data in Fig 7. TG contributed to data in Supplementary Fig S20. TB and LP made intellectual contributions to the design and analysis of the experiments and helped with the drafting of the manuscript.

## Conflict of interest

The authors declare that they have no conflict of interest.

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
