## [Review Process File · The EMBO Journal]

Manuscript EMBO-2014-88740

TDP-1, the *C. elegans* ortholog of TDP-43, limits the accumulation of double stranded RNA

Tassa K Saldi, Peter EA Ash, Gavin Wilson, Patrick Gonzales, Alfonso Garrido-Lecca, Christine M Roberts, Vishantie Dostal, Tania F Gendron, Lincoln D Stein, Thomas Blumenthal, Leonard Petrucelli, Christopher D Link

Corresponding author: Tassa Saldi, University of Colorado

Review timeline:

Submission date:	18 April 2014
Editorial Decision:	02 June 2014
Revision received:	06 August 2014
Accepted:	04 September 2014

Editor: Anne Nielsen

Transaction Report:

1st Editorial Decision

02 June 2014

Thanks a lot for providing us with an outline of the experiments that could be included in the revised version of your manuscript. I have now gone through it in detail in light of the referee reports and also discussed the individual points with a colleague in the editorial team.

I do understand that you would prefer to develop the mechanism of dsRNA regulation by Tdp1 as a separate story; however, I am not fully convinced that the currently offered additional data would suffice for the referees to support publication (since all three referees specifically point to the lack of mechanistic insight as a major concern). An alternative option that may help sway the referees, would be for you to instead include the full data set showing enhanced RNAi in tdp1-depleted animals. Not only would this go well with the genetic data for chemotaxis in tdp1 mutants (and thus address two separate concerns raised by ref#1), it would also add important insight on the functional outcome of Tdp1 depletion, as requested by the referees. I would therefore strongly encourage you to include this data in the revised manuscript.

With regard to the manuscript length, we do not impose limitations here and we would be happy to allow you the required space to develop the study as requested by the referees.

Given the recommendations from the referees - and in light of the experimental outlined supplied - I would therefore invite you to submit a revised version of this manuscript addressing the comments of all three reviewers. In addition to the specific points above this revised manuscript should include all data required to support the technical points raised as well as the three major experiments proposed in your letter to me. I should add that it is EMBO Journal policy to allow only a single round of revision, and acceptance of your manuscript will therefore depend on the completeness of your responses in this revised version.

Thank you for the opportunity to consider your work for publication. I look forward to your revision.

REFEREE REPORTS

Referee #1:

This manuscript describes studies on the role of TDP-1, a *C. elegans* ortholog of human TDP-43, on modulating the transcriptome.

A global analysis of *C. elegans* mutants deleted for *tdp-1* showed that this factor is important to maintain the levels of RNAs with inherent double stranded structure (dsRNA) and to limit nuclear dsRNA accumulation. Apparently, the reduction of dsRNA accumulation seems to be evolutionary conserved in humans, since knockdown of mammalian TDP-43 in HeLa cells and M17 neuronal cells also caused dsRNA accumulation. In addition, TDP-1 seems to influence adenosine-to-inosine RNA editing that it is associated with highly structured regions.

Although the information provided in this manuscript is interesting and the experiments are well designed, the study is mainly descriptive. In fact no mechanism has been explored thoroughly for any of the intriguing questions raised by the output of the study, such as the characteristics of transcripts with increased double-stranded RNA structure in *tdp-1* mutants; the transcript abundance and their splicing pattern in *tdp-1* mutants; the mechanism by which TDP-1 limits dsRNA accumulation and the relevance of dsRNA accumulation to ALS/FTLD pathology.

The authors should in addition address the following topics:

- 1- It is quite striking the contrast between the mild phenotype of *tdp-1* mutant *C. elegans* and the impact of *tdp-1* deletion on dsRNA metabolism/abundance and RNA editing. A possible explanation for this observation should be provided.
- 2- The Authors have observed that many TDP-1 regulated genes with inherent double stranded structures contained either antisense overlaps with another gene or multiple inverted repeats within intronic regions. The Authors should explain clearly if these features are conserved in ortholog human genes.
- 3- The Authors checked the amount and localization of dsRNA in the *C. elegans* TDP-1 mutant by immuno-staining with the dsRNA-specific J2-antibody. Interestingly, these effects seem to be conserved in human cells, where dsRNA accumulation was observed both in nucleus and cytoplasm. However the analysis goes no further and there are no experiments that address the effect of silencing of *tdp-1*/TDP-43 on the levels of other components of the RNAi machinery, in particular of Dicer and Drosha, that are direct interactors of TDP-43 and DGCR8. This question should be answered.
- 4- The Authors find that the most common A-to-I hyperediting in *tdp-1* mutants occurred in intronic regions. They must clarify if the A to I RNA editing can depend on TDP-1-induced changes of *adr-1* and *add-2* levels and if TDP-1 affect expression of *adr* at any level.

Referee #2:

Here the authors perform extensive transcriptomic and bioinformatic analyses of a *C. elegans* strain lacking TDP-1, a homolog of the human disease-linked TDP-43. Although a lot of knowledge has accrued in the past few years about RNA targets of TDP-43, this work identifies interesting and important new information. Initially, deep-sequencing of mRNA transcripts yielded >1700 hits, which did not reveal any common functional denominator. Thus, the authors had the clever idea and also the bioinformatic power to ask if TDP-1 sensitive transcripts were structurally related. Indeed, they found that among the TDP-1 sensitive transcripts, dsRNA was enriched. The authors go on to build on the hypothesis that TDP-1 (and as briefly confirmed in the last figure, human TDP-43) limits the expression of dsRNA and/or mRNA with such structure particularly in intronic regions. Such transcripts tend to be subject of A-to-I RNA editing, as was confirmed as well. Finally, chromatin and RNA immunoprecipitation experiments suggested that TDP-43 acts co-transcriptionally on structured mRNA expression.

This reviewer lacks the bioinformatic expertise to assess the bioinformatic approaches. Within these

caveats, the results and conclusions seem plausible to me. As the authors discuss themselves, it is now necessary to elucidate the molecular details of the claimed co-translational regulation of structured mRNA expression by TDP-43. Moreover, it is desirable to identify human orthologs of the identified TDP-1 sensitive *C. elegans* transcripts, if at all possible, and/or test TDP-43 effects on known human mRNAs with similar double-stranded structure elements in mammalian systems.

Referee #3:

Saldi and colleagues describe in their manuscript a role of TDP-1, the *C. elegans* ortholog of TDP-43, in the formation and/or stability of double-stranded RNA. Using mRNA seq of wild-type and *tdp-1* null animals the authors observe a number of mRNAs with altered expression and processing in the mutant animal. Furthermore the authors find that loss of *tdp-1* results in an accumulation of polyadenylated transcripts with double-stranded RNA structures.

The accumulation of dsRNA in *tdp-1* null and TDP-43 knockdown cells is of broader interest in the context of previous findings by Samarajiwa et al (2009). Samarajiwa and colleagues found that TDP-43 knockdown in mouse neurons resulted in an overexpression of a set of mRNAs that are encoded by type 1 interferon-inducible genes, which specifically respond to cytoplasmic dsRNA.

The finding that TDP-1 plays a fundamental role in controlling dsRNA accumulation and processing is certainly significant and of broad interest, however it is the reviewers opinion that additional experimental evidence is required to support the conclusions. In addition any mechanistic insights how TDP-1 influences the accumulation of dsRNA is missing.

Major issue:

1. A key issue is that the authors rely on the dsRNA-specific J2 antibody to detect dsRNA foci and RIP-seq RNAs with potentially double-stranded regions. Alternative approaches should be used to provide evidence for the accumulation of dsRNA. The Weissman lab recently published an approach for globally monitoring RNA structure in native conditions in vivo with single-nucleotide precision. This method is based on in vivo modification with dimethyl sulphate (DMS), which reacts with unpaired adenine and cytosine residues, followed by deep sequencing to monitor modifications.
2. The authors suggest in their manuscript that ChIP is widely used to determine the association site of RNA binding/processing factors that co-transcriptionally process nascent RNA (Swinburne et al, 2006). The reviewer politely disagrees. CLIP-seq approaches (HITS, iCLIP, or PAR-CLIP) are being used to identify the RNA binding sites of proteins. CLIP-seq inferred TDP-1 binding to intronic sequences, as shown before for TDP-43, would indicate that TDP-1 interacts with nascent RNA. ChIP-seq data only provide evidence that TDP-1 is associated with the respective chromatin regions. Furthermore Table E15 suggests that the TDP-1 ChIP was only performed once.
3. The authors split their analysis of RNA sequences selected by J2-IP in *tdp-1(ok803)* vs wildtype into genes, repeat regions and introns that are differentially enriched. However for a correlation with the TDP-1 ChIP-seq data the authors seem to use only the transcript information ("page 11:") However, we observed a striking enrichment for TDP-1 association among over/under-expressed mature transcripts that contained dsRNA structure as determined by recovery in the J2-IP... Consistent with this idea, analysis of TDP-1 binding in genes with increased dsRNA structure/stability in *tdp-1* mutants (as assayed by J2-IP enrichment) showed about 60% of these genes contained clear TDP-1 binding sites ($p=1.6 \times 10^{-20}$, hgd)". In Figure 6C the authors show an example of TDP-1 binding sites within intronic regions containing excess dsRNA structure/stability along with the J2-IP enrichment of these regions. From this Figure 6C and the described analysis is difficult to infer whether the peak regions selected by J2-IP and the TDP-1 ChIP overlap or simply fall into the same transcript regions.

Minor points:

- Northern or qRT-PCR validations of the mRNA expression changes in wt and *tdp-1* animals is lacking
- likewise exon junction spanning RT-PCR validation of alternative splicing is missing
- it is unclear how reproducible the J2-IP RNA-seq experiments are. A pair-wise correlation of the 3 experiments should be presented.
- What was used as background for the TDP-1 ChIP analysis?
- Figure E10C: is not apparent how many times the RIP-qRT-PCRs were performed and where the primers were positioned.
- Since cellular stress can induced the formation of dsRNA, the author should mention and discuss

the study by Vaccaro et al. (PLOS Genetics 2012), in which the authors describe that TDP-1/TDP-43 Regulates Stress Signaling and Age-Dependent Proteotoxicity in *Caenorhabditis elegans*.

1st Revision - authors' response

06 August 2014

Point by Point responses to reviewer comments:

Reviewer 1 and Reviewer 2

2- The Authors have observed that many TDP-1 regulated genes with inherent double stranded structures contained either antisense overlaps with another gene or multiple inverted repeats within intronic regions. The Authors should explain clearly if these features are conserved in ortholog human genes...//...Moreover, it is be desirable to identify human orthologs of the identified TDP-1 sensitive *C. elegans* transcripts, if at all possible, and/or test TDP-43 effects on known human mRNAs with similar doublestranded structure elements in mammalian systems.

We do not expect a strong conservation in transcript structure between specific *C. elegans* and mammalian orthologs, as much of this structure results from gene arrangements and repeat sequence insertions, which are not well-conserved between invertebrates and mammals. However, we do expect similar *classes* of transcripts (e.g., transposable elements) to show enhanced dsRNA structure as a result of *tdp-1* deletion and TDP-43 knockdown. To test this prediction we have performed a J2-IP in human cell lysate knocked down for TDP-43. We now show several examples of transcripts with increased dsRNA structure by RT-PCR upon reduction of TDP-43 function, including: increased dsRNA structure in the retrotransposon *HervK*, increased dsRNA in a transcript directly bound by TDP-43 (*FUS/TLS*) and increased dsRNA structure in the intronic regions bracketing alternatively spliced exons regulated by human TDP-43 (see **Figure 8**). We have confirmed these results by deep sequencing the RNA precipitate from an independent J2-IP.

Reviewer #1:

1- It is quite striking the contrast between the mild phenotype of *tdp-1* mutant *C. elegans* and the impact of *tdp-1* deletion on dsRNA metabolism/abundance and RNA editing. A possible explanation for this observation should be provided.

This is a valid point raised by this reviewer. One of the major pathways known to counter nuclear dsRNA is A-to-I RNA editing catalyzed by ADAR. In worms, deletion of the functional ADAR ortholog (*adr-2*) also causes no obvious defects, but does result in a chemotaxis defect. This defect is suppressed by genetically removing the RNAi pathway (Tonkin and Bass, *Science*, 2003), indicating that increased dsRNA structure in *adr-2* mutants results in inappropriate metabolism of chemotaxis transcripts by the RNAi pathway. Of import, we now show that *tdp-1* mutant animals have a chemotaxis defect that is also suppressed by genetic removal of RNAi (see **Figure 7**). This result indicates that *tdp-1* mutants have the same phenotypic consequences in worms as deletion of factors known to limit nuclear dsRNA.

3- The Authors checked the amount and localization of dsRNA in the *C. elegans* TDP-1 mutant by immuno-staining with the dsRNA-specific J2-antibody. Interestingly, these effects seem to be conserved in human cells, where dsRNA accumulation was observed both in nucleus and cytoplasm. However the analysis goes no further and there are no experiments that address the effect of silencing of *tdp-1*/TDP-43 on the levels of other components of the RNAi machinery, in particular of Dicer and Drosha, that are direct interactors of TDP-43 and DGCR8. This question should be answered.

We assume that this reviewer is asking this question because both of these enzymes metabolize dsRNA and decreased activity of these proteins in *tdp-1* mutants could cause increased dsRNA accumulation. This is an excellent suggestion. Examination of transcript levels for both dicer and drosha (as well as all RNAi pathway factors in *tdp-1* mutants indicates that the levels of these transcripts are not significantly changed

compared to wild type (see **Expanded View Table 1**). Further, suppression of *tdp-1* mutants chemotaxis defect by genetically removing the RNAi pathway indicates RNAi is functioning in *tdp-1* mutants and increased dsRNA is likely up regulating the RNAi pathway. As we did not identify miRNA precursors as having increased dsRNA structure in *tdp-1* mutants, we feel that there is no support for pursuing changes in the miRNA pathway in *tdp-1* mutants, at least as it relates to dsRNA accumulation.

4- The Authors find that the most common A-to-I hyperediting in *tdp-1* mutants occurred in intronic regions. They must clarify if the A to I RNA editing can depend on TDP-1-induced changes of *adr-1* and *adr-2* levels and if TDP-1 affect expression of *adr* at any level.

This is a valid point by this reviewer. We have examined the worm ADAR genes in our transcriptome sequencing and found no change in transcript abundance or splicing of either *adr-1* or *adr-2* (see **Figure E7**). Unfortunately there is no available antibody directed against ADR-2 in worms so we are not able to determine protein levels of ADR-2 in *tdp-1* mutants. However, we have confirmed that ADR-1 protein levels are not substantially changed in *tdp-1* mutants (**Figure E7b**). These results support the conclusion that *tdp-1* does not limit A-to-I RNA editing by controlling the expression/processing of ADAR transcripts or ADR-1 protein.

Reviewer 2:

As the authors discuss themselves, it is now necessary to elucidate the molecular details of the claimed co-translational regulation of structured mRNA expression by TDP-43.

We agree with reviewer 2 that the next important step in this line of research is to understand how TDP-1/TDP-43 limits dsRNA. In response to this request we have assayed recombinant human TDP-43 for RNA chaperone activity. As now shown in **Figure 9**, we found that TDP-43 displays strand displacement activity a wellcharacterized activity of the known RNA chaperone, human La protein. While we cannot definitively conclude that TDP-43 limits dsRNA in all affected transcripts by acting as an RNA chaperone, this result provides a plausible molecular mechanism of how TDP-43 may limit dsRNA in vivo. Namely, by acting as a cotranscriptional RNA chaperone that maintains appropriate nascent RNA structure.

Reviewer 3:

1. A key issue is that the authors rely on the dsRNA-specific J2 antibody to detect dsRNA foci and RIP-seq RNAs with potentially double-stranded regions. Alternative approaches should be used to provide evidence for the accumulation of dsRNA. The Weissman lab recently published an approach for globally monitoring RNA structure in native conditions in vivo with single-nucleotide precision. This method is based on in vivo modification with dimethyl sulphate (DMS), which reacts with unpaired adenine and cytosine residues, followed by deep sequencing to monitor modifications.

We agree that the Weissman lab study is very powerful and elegant. However, this protocol was done on isolated cells (yeast and human) and not in a whole animal. Therefore, we feel the amount of time it would take to perfect this protocol for use in a whole animal system is prohibitive and could ultimately not work efficiently (or at all) in *C. elegans*. However, the J2 antibody has been extensively characterized to bind only dsRNA and this association has been described on the atomic level (*Schonbron et al, 1991 and Bonin et. al, RNA, 2000*). J2 antibody has been used in numerous studies for both immuno-detection of dsRNA in whole cells/animals as well as for immunoprecipitation of dsRNA in lysate made from whole tissue (Richardson etl al, 2010; Weber et. al, 2006; White et. al, 2014; Kaneko, 2011). We also demonstrate in this manuscript that the J2 antibody immunostaining (**Figure 2**) and IP specificity (**Figure E9**) are abolished with treatment of a dsRNA specific RNase (V1). Further, we have now produced a transgene expressing a dsRNA binding domain fused to GFP and show that the GFP signal tightly overlaps with the J2 signal, (see **Figure E5**). We feel that this data adequately demonstrates the specificity of the J2 antibody. In addition,

experiments now presented in **Figure 9** show that TDP-43 can act as an RNA chaperone, providing support for TDP-43 affecting RNA structure independent of the J2 antibody.

2. The authors suggest in their manuscript that ChIP is widely used to determine the association site of RNA binding/processing factors that co-transcriptionally process nascent RNA (Swinburne et al, 2006). The reviewer politely disagrees. CLIP-seq approaches (HITS, iCLIP, or PAR-CLIP) are being used to identify the RNA binding sites of proteins. CLIP-seq inferred TDP-1 binding to intronic sequences, as shown before for TDP-43, would indicate that TDP-1 interacts with nascent RNA. ChIP-seq data only provide evidence that TDP-1 is associated with the respective chromatin regions. Furthermore Table E15 suggests that the TDP-1 ChIP was only performed once. We agree with reviewer #3 that CLIP (along with other proposed protocols) can be a useful approach for demonstrating direct association of TDP-1 with introns of specific transcripts; this is why we undertook the UV-CLIP RT-PCR experiments shown in **Figure E15**. These experiments demonstrate unequivocally that TDP-1 associates with transcript RNA for a subset of genes we targeted. However, for most of the target transcripts, TDP-1 overexpression was required to demonstrate TDP-1 binding. We interpret this observation as evidence that CLIP (at least in our model system) has limited sensitivity, which is why we pursued the TDP-1 ChIP-seq.

Reviewer #3 also argues that chromatin immunoprecipitation (ChIP) is not the appropriate method to determine sites of TDP-1 association in nascent RNA. It is further argued that because previously published TDP-43 binding sites fall in intronic regions, this is evidence that TDP-43 associates with nascent mRNA. While a binding site located within an intron is suggestive, it is not conclusive proof that TDP-43 binds nascent mRNA and cannot show that TDP-43 associates with RNA cotranscriptionally. TDP-43 could be binding retained introns in otherwise processed transcripts that are far from the transcription site. Alternatively, TDP-43 could be associating with free introns that are already spliced. Given the published data showing that TDP-43 toxicity is suppressed by mutations in debranching enzyme (Dbr1) (Armakola et. al, 2012), the possibility that TDP-43 binds free introns is a viable possibility. Therefore, determining whether TDP-43 binds cotranscriptionally to nascent RNA (and which RNAs) versus binding a processing intermediate is crucial to understanding the function of this protein. The most widely used protocol to show cotranscriptional (nascent mRNA) association is ChIP. ChIP has been previously used to identify binding sites in the RNA of a wide spectrum of RNA processing factors that act cotranscriptionally including; mRNA 5'-end processing factors (Brannen et. al, Molecular Cell, 2012 and Zhifa et. al, Genes and Development, 2010); mRNA 3'-end processing factors and RNA degradation factors (Glover-Cutter et. al, Nature Structural Molecular Biology, 2008 and Komarnitsky et. al, Genes and Development, 2002); splicing factors (Görnemann et. al, Molecular Cell, 2005) and micro-RNA processing factors (Wagschal et. al, Cell, 2012). In addition, sites of association by another RNA binding protein known to be involved in ALS, Fus/TLS, have been characterized using ChIP (Morlando et. al, EMBO, 2012 and Schwartz et. al, Genes and Development, 2012).

However, we do agree that it is possible that some of the identified TDP-1 ChIP peaks could be due to association with the DNA as TDP-43 is capable of binding both RNA and DNA. In order to address this concern we have repeated the TDP-1 ChIP-seq in both the presence and absence of RNase and now show that the majority (if not all) of TDP-1 ChIP peaks are abolished in the absence of RNA. This experiment shows conclusively that TDP-1 ChIP peaks represent cotranscriptional association with RNA. TDP-1 ChIP-seq has now been performed twice, and this fact is clarified in the text and methods.

3. The authors split their analysis of RNA sequences selected by J2-IP in *tdp-1(ok803)* vs wildtype into genes, repeat regions and introns that are differentially enriched. However for a correlation with the TDP-1 ChIP-seq data the authors seem to use only the transcript information ("page 11:") However, we observed a striking enrichment for TDP-1 association among over/under-expressed mature transcripts that contained

dsRNA structure as determined by recovery in the J2-IP... Consistent with this idea, analysis of TDP-1 binding in genes with increased dsRNA structure/stability in *tdp-1* mutants (as assayed by J2-IP enrichment) showed about 60% of these genes contained clear TDP-1 binding sites ($p=1.6 \times 10^{-20}$, hgd)". In Figure 6C the authors show an example of TDP-1 binding sites within intronic regions containing excess dsRNA structure/stability along with the J2-IP enrichment of these regions. From this Figure 6C and the described analysis is difficult to infer whether the peak regions selected by J2-IP and the TDP-1 ChIP overlap or simply fall into the same transcript regions.

As indicated on **page 6** of the main text, we performed the J2-IP on *in vivo* extract so if a transcript contains a stretch of 40bp or more of dsRNA, the entire transcript will be selected in the J2-IP, not just the double stranded region. Therefore, we are not able to determine the exact location of the double stranded section, but only that the transcript contains double stranded structure. On **page 12** we also indicate that TDP-1 ChIP peaks are enriched in both genes with increased structure in the mutant as well as introns with increased structure in the mutant. For introns we were able to locate the ChIP peak directly to the J2-IP enriched intron as most pre-mRNAs isolated by J2-IP only retained the structured intron. To further address this concern, we have also compared the TDP-1 ChIP peaks to locations in the genome that contain annotated repetitive elements (such as transposons). As we now indicate on **page 12**, 51% of TDP-1 ChIP peaks overlapped with these regions indicating TDP-1 associates with potentially structured RNAs.

Minor points:

- Northern or qRT-PCR validations of the mRNA expression changes in wt and *tdp-1* animals is lacking

We have done this and data is shown in **Figure E2**.

- likewise exon junction spanning RT-PCR validation of alternative splicing is missing

We have now done this and data is shown in **Figure E3 and Figure 6**.

it is unclear how reproducible the J2-IP RNA-seq experiments are. A pair-wise correlation of the 3 experiments should be presented.

We have now done this and data is shown in **Figure E8**.

- What was used as background for the TDP-1 ChIP analysis?

The RNase treated ChIP-seq is now used as the background for the ChIP analysis.

- Figure E10C: is not apparent how many times the RIP-qRT-PCRs were performed and where the primers were positioned.

These experiments were performed on two biological replicates. We thank the reviewer for pointing out our omission and we will update the text. **Figure E10C (now Figure E15)** indicates the intron that was tested for TDP-1 association and the actual primer sequences are included in **Table E13**.

- Since cellular stress can induced the formation of dsRNA, the author should mention and discuss the study by Vaccaro et al. (PLOS Genetics 2012), in which the authors describe that TDP-1/TDP-43 Regulates Stress Signaling and Age-Dependent Proteotoxicity in *Caenorhabditis elegans*.

We have now added the following sentence to the discussion:

Vaccaro et. al. demonstrated that the *tdp-1(ok803)* allele renders worms hypersensitive to oxidative stress and interacts with the *C. elegans* insulin-like signaling (ILS) pathway. Interestingly, *daf-16* (the key transcription factor controlling ILS in *C. elegans*) is hyper-edited in *tdp-1(ok803)* (Figure 3C), suggesting increased dsRNA structure in *tdp-1(ok803)* worms could contribute to the altered stress sensitivity.

Thank you for submitting your revised manuscript for The EMBO Journal. Your study has now been seen by the original referees (comments included below) and as you will see they find that all criticisms have been adequately addressed. I am therefore happy to inform you that your manuscript has been accepted for publication with us.

If you have any questions, please do not hesitate to contact me. Thank you for your contribution to The EMBO Journal and congratulations on this nicely executed work.

REFEREE REPORTS

Referee #1:

The authors have given reasonable replies to this referee's queries. The manuscript is now suitable for publication

Referee #2:

The authors have responded well to the reviewer comments.

When processing the final text file, please remember to correct the annotations to the new Figure 8E (not A) on page 14, lines 11 and 19.

Referee #3:

All the reviewer's concerns were adequately addressed.

An accession number for the submitted sequencing data should be indicated in the manuscript.